# Balanced imitation sustains song culture in zebra finches

Ofer Tchernichovski [1,2✉], Sophie Eisenberg-Edidin[1,3] & Erich D. Jarvis [3,4✉]

Songbirds acquire songs by imitation, as humans do speech. Although imitation should drive convergence within a group and divergence through drift between groups, zebra finch songs sustain high diversity within a colony, but mild variation across colonies. We investigated this phenomenon by analyzing vocal learning statistics in 160 tutor-pupil pairs from a large breeding colony. Song imitation is persistently accurate in some families, but poor in others. This is not attributed to genetic differences, as fostered pupils copied their tutors' songs as accurately or poorly as biological pupils. Rather, pupils of tutors with low song diversity make more improvisations compared to pupils of tutors with high song diversity. We suggest that a frequency dependent *balanced imitation* prevents extinction of rare song elements and overabundance of common ones, promoting repertoire diversity within groups, while constraining drift across groups, which together prevents the collapse of vocal culture into either complete uniformity or chaos.

[1] Department of Psychology, Hunter College, New York, NY, USA. [2] Neuroscience Program, The CUNY Graduate Center, New York, NY, USA. [3] Laboratory of Neurogenetics of Language, The Rockefeller University, New York, NY, USA. [4] Howard Hughes Medical Institute, Chevy Chase, MD, USA. ✉email: tchernichovski@gmail.com; ejarvis@rockefeller.edu

Vocal culture is the cornerstone of spoken language, but is not unique to humans[1–5]. Like humans, songbirds acquire their vocal repertoire via imitation (i.e., vocal learning)[6–9], a process that can give rise to local dialects that persist over hundreds of generations[10,11]. However, the repertoire of vocal learning birds also has a strong genetic component[11–13]. Across populations, innate biases in song perception, production, and learning sustain species-specific song repertoires[13–16]. Canaries, for example, will faithfully imitate songs of abnormal combinatorial structure, but later, as they reach maturity, alter their songs to match a species typical song syntax to which they have not been exposed[17]. Similarly, zebra finch males (females do not sing) that are trained with random combinatorial transitions of syllable types will generate combinations that are biased toward the species typical[18,19]. Innate biases may unfold at the scale of generations, too; the descendants of isolated zebra finch tutors, who produce aberrant songs, produce increasingly species typical songs[2,19].

Theoretically, vocal imitation should drive song repertoire convergence within groups and divergence across groups[20–22]. Meanwhile, innate biases in imitation might constrain drift. In reality, however, zebra finch songs remain highly diverse within groups and vary only mildly across them[22]. We do not know whether this diversity serves any function in domesticated zebra finches, but high similarity between songs could potentially generate impoverished communication systems that convey little information about individual identity[23,24]. In wild songbirds, across species, and even subspecies, the magnitude of individual song variability differs strongly, often for no apparent reason. For example, the songs of the wild Australian zebra finch (*Taeniopygia guttata castanotis*) are much more variable among individuals than those of the closely related wild Timor zebra finch (*Taeniopygia guttata guttata*)[25]. This variability persists despite the fact that they live in similar climates and have similar social organization.

Here we test how a rich polymorphic repertoire of song syllables is sustained during cultural transmission[26] in the Australian zebra finch. We quantify song polymorphism using novel measures of vocal states and acoustic diversity, for studying the statistics of song imitation in a large colony. We find that the polymorphic repertoire is sustained by pupils spontaneously increasing song diversity when tutors have low-diversity songs, and imitating with greater fidelity when tutors have high-diversity songs, a process we call balanced imitation.

## Results

We recorded the songs of 160 zebra finch tutor–pupil pairs (68 tutors and 160 pupils; 228 birds overall) at the Rockefeller University Field Research Center colony, which consisted of over 800 birds during the 1-year period of recording. Of the 160 pupils, 130 pupils were housed with their biological parents, and 30 pupils with foster parents. We also analyzed song imitation across three generations including 14 grand-tutors and 35 grand-pupils. All birds were housed in individual breeding cages with parents (either biological or foster) and other offspring, and kept visually isolated from adjacent breeding cages. With this social regimen, we found no evidence of song imitation across families (Supplementary Fig. 1). From each bird, we recorded undirected songs (produced in isolation) for over a week to obtain a sample of at least 1000 song syllables per bird. Directed songs to females were also recorded, but not analyzed for this study.

**Imitation outcome varied across families.** We first measured similarity[27] between tutor and pupil songs based on acoustic features (pitch, frequency modulation, Wiener entropy, and spectral continuity)[28]. We observed considerable variability in the distribution of song similarities between pupils and their individual tutors (mean = 69%; range 20–100%; CV = 0.28, Fig. 1a). To test for family influence, we identified 24 families that had multiple clutches with males, calculated the mean song similarity between pupils and their tutors of each clutch, which allowed us to normalize out the effect of song convergence between siblings[20]. We then calculated the coefficients of variance across clutches within families and compared it to the coefficient of variance across families (Fig. 1b). We found that imitation similarity was much more variable across families than within families (Kruskal–Wallis chi-squared = 44.727, df = 23, p-value = 0.006).

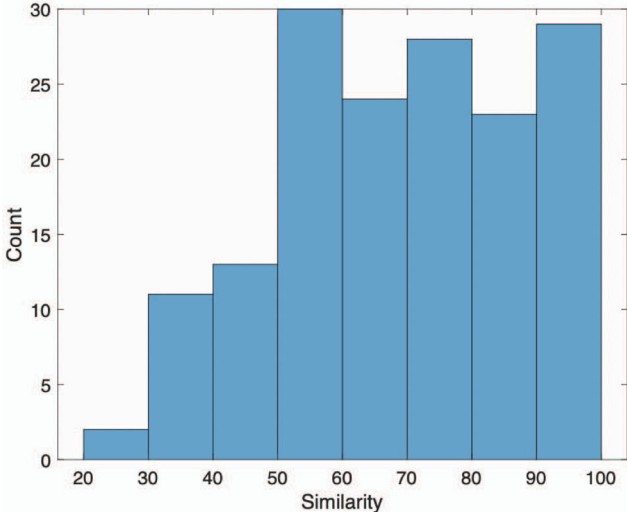

### a. song similarities between pupils and tutors

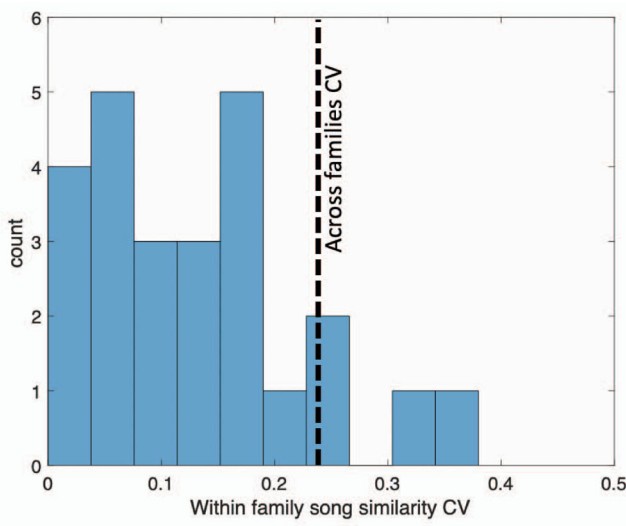

### b. song similarities within families

**Fig. 1 Distribution of song similarity between pupils and their tutors. a** Histogram of song similarities between 160 pupils and their tutors. **b** Analysis of variance in song similarity between and within families. Data include 24 families with more than one clutch with males. Similarity scores were averaged within clutch members and coefficient of variance (CV = 0.14 ± 0.02) of similarity scores were calculated across clutches. CV of the same data (averaged within clutches) across families = 0.24 is presented as a dotted line. Source data for this figure is in Supplementary Data File 1.

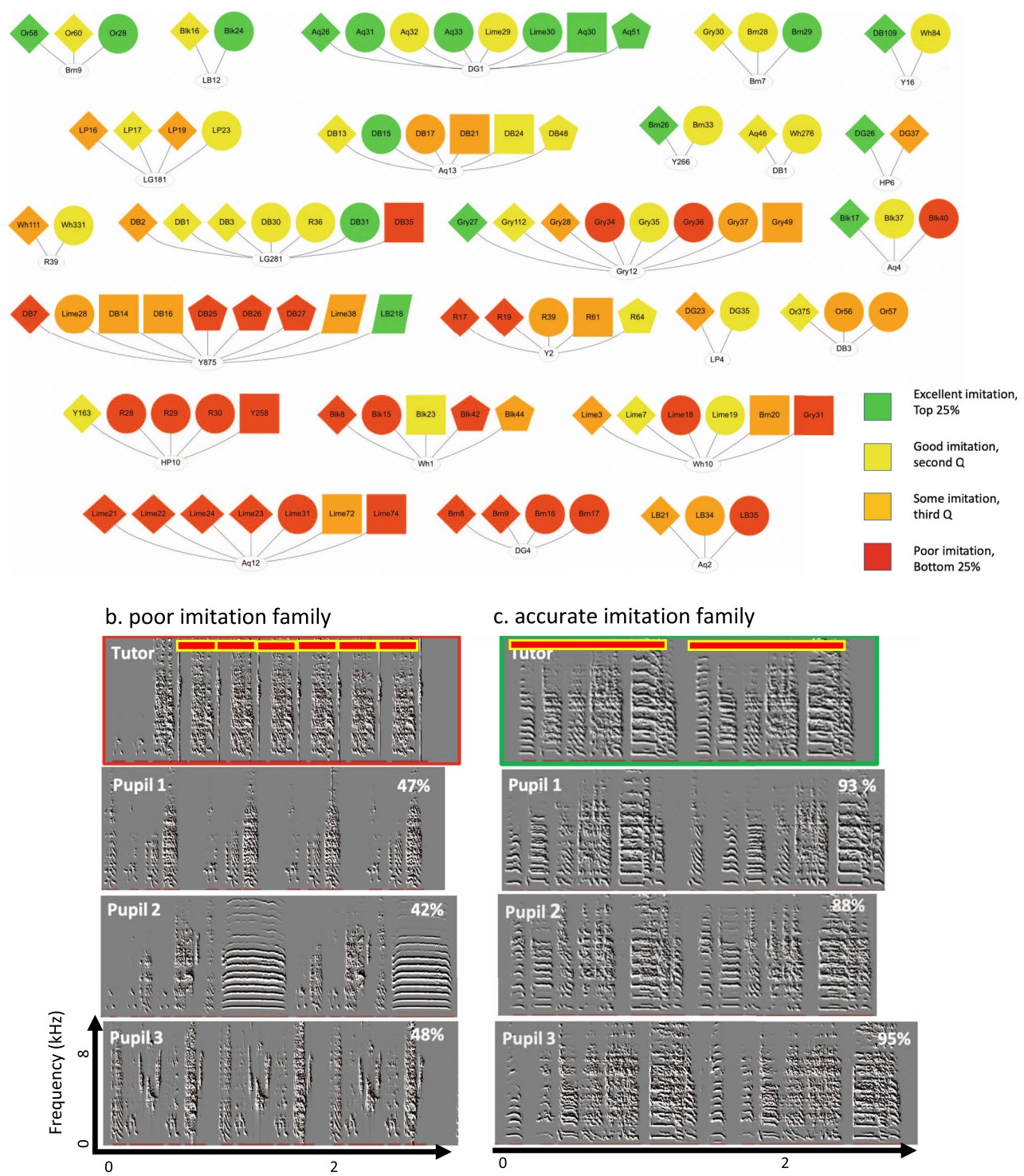

**Fig. 2 Imitation outcome varies across families. a** 24 song tutoring lineages. All tutors had pupils in more than a single clutch. Each node represents one individual animal. Node shape represents pupils from the same clutch. Tutor nodes are presented on the bottom and pupil nodes on the top. Similarity scores are presented as quartiles (green for best imitations and red for poorest). Lineages are sorted according to the mean similarity between tutor and pupils from highest (top) to lowest (bottom). **b**, **c** Examples of song imitations from tutor AQ12 with a low similarity family (**b**) and from tutor DG1 with a high similarity family (**c**). Imitation outcomes are presented as percent acoustic similarity estimates on each sonogram. Red bars outline the repeated song motifs of the tutors. Source data for this figure is in Supplementary Data File 1.

In certain families, across clutches, song imitation tended to be almost exclusively accurate (top quartile), in some modest (middle quartile), and others generally poor (Fig. 2a). To assess whether this variance in imitation outcome was genetic, we compared song imitation between biological and foster pupils. Foster pupils imitated their tutor as well as biological ones (biological similarity: 68.2 ± 1.7%, $n = 130$; foster similarity: 70.0 ± 3.6%, $n = 30$, mean ± S.E.M. hereafter). Therefore, the variability we observed in imitation outcomes across families cannot be explained by genetic variability. Instead, we noted that variability in imitation among pupils appeared to be associated with tutor song structure. For example, tutor Aq12 had a very simple song with one syllable-type containing two notes and none of his pupils imitated this syllable or song accurately. Instead, some pupils introduced apparently novel syllable types not found in the tutor in developing their own songs (Fig. 2b). In contrast, tutor DG1 had a more complex song, with five syllable types containing six notes, and all of his pupils imitated the syllables and the sequence much more accurately, with little to no introduction of novel syllables (Fig. 2c). In both cases, pupils still produced their syllables in repeated song motifs of 2–6 syllable types, as is typical of zebra finches (Fig. 2b, c). This suggested to us that pupils might more accurately imitate tutor songs that are rich in acoustic structure (i.e., acoustically diverse), while improvising upon impoverished tutor songs.

**Syllable-type diversity is not correlated between tutor and pupil songs.** If this impoverished tutor song hypothesis were true, we would expect to find that as tutor syllable diversity decreases, pupil' imitation similarity also decreases; conversely, we would expect to see biases in the correlation between tutor syllable diversity and pupil syllable diversity at extreme ranges of tutor diversity. To test this hypothesis quantitatively, we sought a measure of syllable acoustic and syntax diversity. We selected a random group of 80 adult tutor–pupil pairs, and segmented their songs into syllable units using an amplitude threshold[27]. Song syllables were automatically clustered into types based on their acoustic features (Fig. 3a, b)[27]. We then calculated the relative frequency (abundance) of each syllable-type and used Shannon information entropy[27] to measure syllable acoustic diversity produced by each bird. Specifically, for each bird's song, we calculated the proportion $p_i$ of syllables produced for each syllable-type $i$, and computed entropy as $-\sum p_i(\log_2(p_i))$. The measure weighs each vocal element (syllable) by its abundance, and presents the entropy (diversity) of the distribution in units of bits. We used the same Shannon information measure to also evaluate syllable transition diversity (song-syntax entropy[29]). The Shannon information entropy has limited bearing on capturing combinatorial complexity[30], but it is a better estimate of diversity compared to just counting syllable types because it considers the frequencies (abundances) of each type. The more syllable types produced, and the more even their abundances are, the higher the entropy.

The distribution of syllable-type diversity of songs in the population was asymmetric, with most songs in the range of 2.5–3 bits and a left tail of rare songs with low syllable diversity (Fig. 3c). Surprisingly, there was no statistically significant correlation between tutor and pupil syllable diversity ($R^2 = 0.079$, NS). Looking separately at pupils who imitated above (and below) average showed no correlations either (Fig. 3d). Further, there was no correlation between tutor syllable diversity and acoustic similarity between tutor and pupil songs (Fig. 3e). To better estimate how tutor syllable diversity may affect the cultural transmission, we calculated song acoustic similarity in reverse, from pupil to tutor. We call this a measure of "influence" because

it tells us how much of the pupil's song is influenced by the tutor. However, influence in pupils was not significantly correlated with tutor song (Fig. 3f). Near zero correlations were also observed for song-syntax (bigram) transitions between pairs of syllable types (Supplementary Fig. 1). In sum, our syllable-type diversity measure failed to capture any aspect of song learning, nullifying all our attempts to evaluate our impoverished tutor song hypothesis.

**Half of the pupils recombine syllables.** Puzzled by the lack of even a weak correlation between tutor and pupil syllable and syntax diversity, we examined cases of most accurate imitation. We found frequent inconsistencies, as is typical of zebra finches in the boundaries of corresponding syllables in the songs of tutors and their pupils, even in cases of accurate imitation. This was not primarily due to measurement (segmentation) errors, but because pupils often modified or recombined the units they imitated (Fig. 3g). We assessed a lower bound estimate of similarity in the syllable boundaries of tutor and pupil songs, restricting the analysis to those syllables whose acoustic structure was clearly and fully imitated (either as a single unit or in parts) by the pupil (examples in Fig. 3g–k). With this strict criterion, analysis of syllable imitations in 33 randomly selected tutor–pupil pairs revealed modification of syllable boundaries in 47 cases (22%) of the copied syllables. Overall, 54% (18/33) of the pupils showed at least one case of altering syllables units. Interestingly, all 47 cases were of merging tutor syllables, rather than splitting. However, splitting might be more difficult to detect, and if so, our analyses would be an underestimate of the magnitude of syllable recombination (see "Methods" section).

**Vocal state measures capture balanced imitation.** Given the extent of syllable recombination, we next sought an alternative quantitative measure that captures acoustic diversity at the sub-syllabic level, which would be, by design, insensitive to syllable recombination. For each of the 160 tutor–pupil pairs, we calculated continuously (in 10 ms FFT windows excluding silences, but without segmentation) three acoustic feature vectors: pitch, Wiener entropy (width of power spectrum), and frequency modulation[28]. Histograms of these features for all birdsongs in our sample reveal several concentrations, and we used the contours of these concentrations to partition the entire acoustic space of the songs into 10 regions. To visualize these concentrations, we present 2D slices of the feature space according to four peaks in the distribution of pitch (Fig. 4a), which we labeled very low, low, medium, and high. These four slices show distinct concentrations of the 10 regions, that we will call vocal states (Fig. 4b). The two concentrations in the highest and lowest pitch regions consisted of down-modulated and up-modulated sounds, respectively (vocal states 1 and 2, for lowest pitch, and 9 and 10 for highest pitch). The two central pitch regions (low and medium) consisted of similar types of vocal states, and two additional states (4 and 7) centered at zero frequency modulation represent non-modulated harmonic sounds. With the vocal states of the population categorized, we can consider each song as a long sequence of vocal states, calculated in small (10 ms) time windows. We next analyzed the distribution of vocal states, by calculating the relative abundances of sounds within each vocal state for each bird.

Similar to syllable acoustic and syntax diversity, for each bird's song, we calculated acoustic diversity over the 10 vocal states using Shannon information entropy[27], $-\sum p_i(\log_2(p_i))$, but here $p_i$ is the proportion of sounds within each vocal state $i$. Hereafter we will refer to this measure as "song diversity". The highest theoretically possible diversity, with a uniform distribution of the 10 vocal states, is 10 times $-0.1(\log_2(0.1)) = 3.32$ bits. Pooling

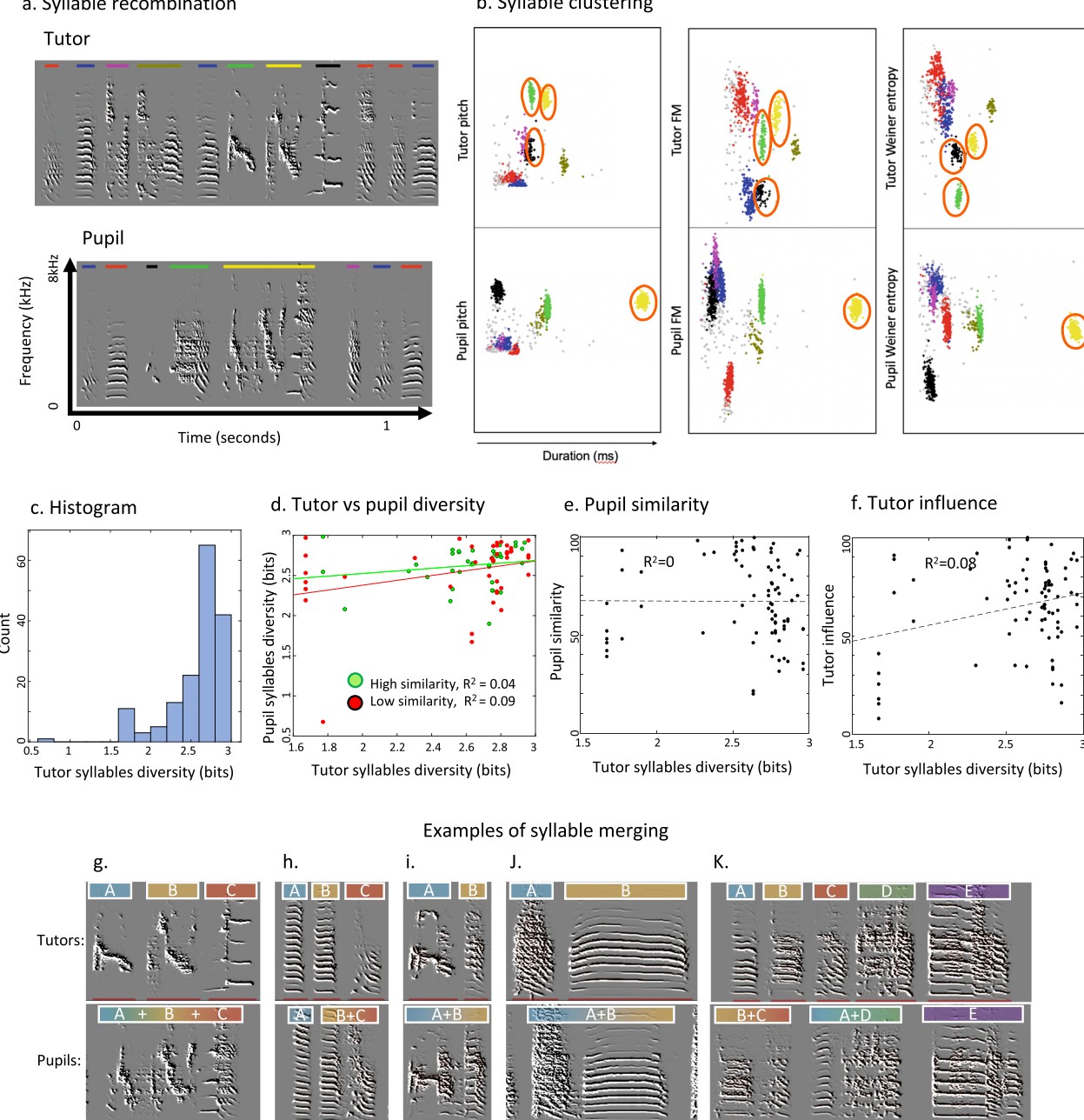

**Fig. 3 Syllable-type diversity. a** Example sonograms of a tutor–pupil song pair. Syllable types are color-coded by lines above them. Color lines above each syllable indicate clusters computed separately for tutor and pupil in **b**. Note that syllable types are bird specific and color codes have no correspondence between tutor and pupil, e.g., green, yellow, and black labeled syllable types in tutor song merged into a single type (yellow labeled) in pupil's song. **b** 2D scatter plots of syllable acoustic features: duration versus mean pitch, mean frequency modulation (FM), or mean Wiener entropy (a measure of the width of the power spectrum). The color of each marker indicates its computed syllable-type (type = cluster in feature space). Colors of clusters correspond to syllable-type colors shown in **a**. **c** Histogram of syllable-type diversity, pooled across all birds. **d** Regression analysis between tutor and pupil syllable-type diversity, showing no significant correlation for pupils with high or low imitation similarity of their tutors. **e**, **f** Tutor syllable diversity is not correlated with pupil song imitation similarity (**e**), or influence of tutors on pupils (**f**). **g–k** Examples of five tutor–pupil pairs with syllable recombination, namely merging in pupil songs. Source data for this figure is in Supplementary Data File 1.

vocal states across all 228 birds recorded gives a diversity of 3.24 bits. This could mean that either song diversity tends to be high within-subjects, or alternatively, that different song "morphs" are evenly distributed. In the latter case, song diversity would be low within-subjects and high when pooled together. Interestingly, the median song diversity of individual birds in our population was 3.14 bits, fairly close to the pooled diversity and to the upper theoretical limit, suggesting a trend to develop acoustically "balanced" songs with respect to the 10 vocal states. The distribution of song diversity was asymmetric with a longer left tail (Fig. 4c) going down to about 2.3 bits, which would be equivalent to a song containing only 5 vocal states. This distribution of song diversity was stable during the lifetime of the colony (Supplementary Fig. 2).

Unlike the syllable diversity measures, tutor and pupil song diversity estimates based on vocal states were positively and

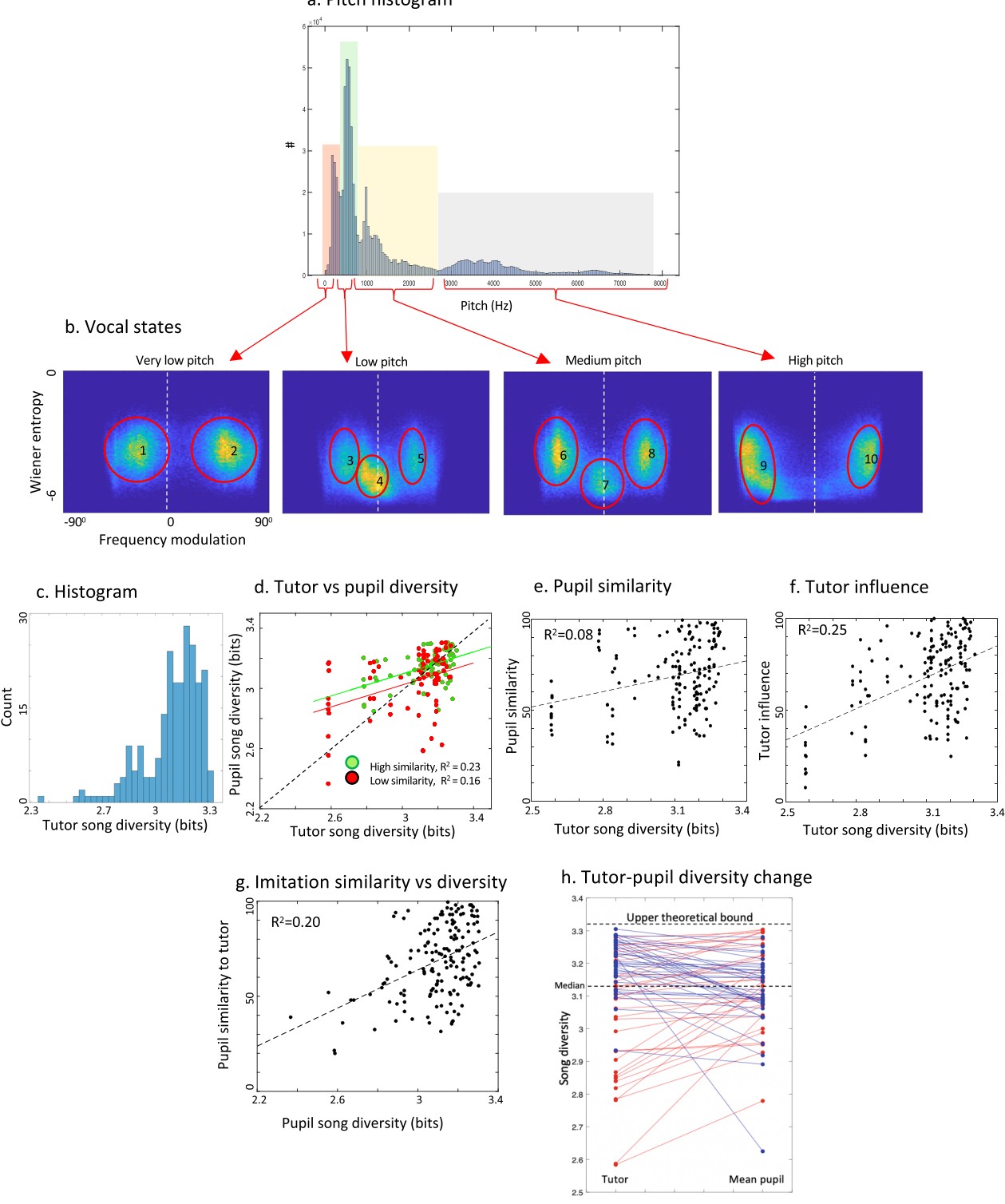

**Fig. 4 Vocal states and diversity in zebra finch songs. a** Histogram of pitch, calculated in 10 ms windows and pooled for all songs[31]. Shadings show partitioning into four regions according to contours of the pitch distribution. **b** Two-dimensional heatmaps of frequency modulation and Wiener entropy for each of the four-pitch regions. Red circles outline 10 clusters around which vocal states are defined. **c** Histogram of song diversity for all male birds recorded. **d** Song diversity in tutor songs versus pupil songs. Colors show $R^2$ separately for high and low similarity birds. **e** Tutor song diversities vs. similarity with pupil songs ($R^2 = 0.08$, $t = 1.9$, Linear mixed-effects model NS). **f** Tutor song diversities vs. the influence of tutor song on pupils ($R^2 = 0.25$, Linear mixed-effects model $t = 4.8$, $p = 4.2e−6$). Vertically aligned markers are often birds from the same lineage. The trend remains significant after removing the lowest diversity families (<2.7, $p = 0.003$). **g** Pupil song diversity vs. pupil song similarity to tutors. **h** Song diversity in families, comparing the songs of tutors versus the mean diversity across all of their pupils' songs. The upper dotted line represents the upper theoretical bound where all vocal states are equally abundant. The lower dotted line indicates the median. Blue lines show decreases, and red shows increases in pupil song diversity. Source data for this figure is in Supplementary Data File 1.

significantly correlated (Fig. 4d, $R^2 = 0.21$, linear mixed-effect model: $n = 160$ pairs, estimate $= 0.40$, $t = 5.59$, df $= 159$, $p = 9.5e-8$). As expected, the correlation with tutor' song diversity explained only a fraction (21%) of the variance observed in pupil song diversity. In cases of poor imitation (below median, red markers), the correlation was driven primarily by tutors of low song diversity. However, all belonged to a single branch in our colony family tree. This may call into question the validity of the correlation observed in lower quartiles of pupil imitation similarity. However, in the top two quartiles of pupil imitation similarity the coefficient of determination is similarly robust ($R^2 = 0.23$, Fig. 4d) with no apparent interaction with specific family branches.

As with syllable-type diversity, pupil imitation similarity was not significantly affected by tutor' vocal state song diversity (Fig. 4e, $R^2 = 0.08$, NS). However, tutor' song diversity was strongly and significantly correlated with influence (Fig. 4f, $R^2 = 0.25$, linear mixed-effect model: $n = 160$ pairs, estimate $= 50.5$, $t = 4.76$, $p = 4.2e-6$). Note that influence tells us how much of a pupil's song is influenced by his tutor. For example, an imitation ABC->ABCDEF will give us 100% imitation similarity because all of the tutor's sounds are present in pupil's song, but only 50% influence because half of the pupil's song is improvised. Indeed, for tutors with high song diversity, the diversity of pupil' songs is centered on the diagonal (identity) line (Fig. 4d). However, for tutors with low song diversity, pupil song diversity, in most cases, above the diagonal (Fig. 4d). For example, out of 34 tutor songs with diversity below 3, only 5 pupil songs are below the diagonal. That is, pupils of tutors with low song diversity often imitated them, but were less influenced by them: they often made additions that increase song diversity. These "low influence" songs did not resemble neighboring birds' songs (Supplementary Note 1). We, therefore, suspect that these additions are improvisations, namely they are likely to be either modified versions of tutor song elements, or innate syllable types.

Assuming a natural trend to develop high-diversity songs either via imitation or improvisation, we wondered why songs of low diversity were not rarer in our colony. We tested which factors may sustain songs of low diversity across generations and found that pupils that imitated poorly, regardless of tutor song diversity, tended to have low-diversity songs (Fig. 4g, $R^2 = 0.20$, $t = 5.7$, $p = 6.2e-8$). To directly test for interaction between imitation accuracy and song diversity, we ran a linear mixed-effect model to explain pupil song diversity with two fixed effects: the diversity of the tutor song, and the acoustic similarity to the tutor song (how much of it was copied). Results confirmed that both factors contribute about equally to pupil song diversity (imitation similarity: $t = 5.0$, $p = 1.4e-6$; tutor song diversity: $t = 4.6$, $p = 7.9e-6$).

In sum, although our syllable diversity measure failed to capture any relationship with song imitation, bypassing syllable recombination by measuring song diversity based on vocal states (without segmentation) revealed two effects: First, low diversity in a tutor's song was not associated with lower imitation similarity in the pupil but with lower influence on the pupil, indicating a tendency in pupils to increase song diversity, which we call "balanced imitation". Second, low diversity in a pupil's song (but not in tutor song) is associated with poor imitation similarity in the pupil. Together these effects can explain the stable polymorphism in song diversity across generations: on the one hand, pupils tend to increase song diversity when tutored by a low-diversity song model, but on the other hand, poor imitation is associated with a decrease of song diversity in pupils' songs. Consistent with this interpretation, when we plotted song diversity of each tutor against the mean song diversity of all of his pupils, the mean song diversity in pupils of low-diversity

(below median) tutors was often higher than that of their tutors, and vice versa (Fig. 4h). That is, despite the overall positive correlation between tutor and pupil song diversity, we see frequent reversals such that a large proportion (42%) of pupils with low song diversity had tutors with high (above median) song diversity, and vice versa.

**Balanced imitation across multiple generations**. We further explored reversals across multiple generations, and analyzed 14 family branches, where we had song imitation data across two generations of pupils. We found that in the families where the first-generation pupils imitated poorly, there was often some recovery in imitation accuracy in the second-generation, the grand-pupils (Fig. 5a). For example, in the two lineages (HP10 and DG4) with the greatest number of first-generation pupils that imitated poorly, all of the second-generation (grand) pupils imitated the song of their tutor more accurately than the tutor's imitation of the grand tutor. Sonograms revealed that, in both lineages, the grand-tutor songs were unbalanced: Tutor HP10 had a very high-pitched song (Fig. 5b), whereas tutor DG4's song included numerous harmonic stacks (Fig. 5c). In both cases, their pupils developed songs that appear to be more acoustically "balanced," and ones that the grand-pupils imitated accurately (Fig. 5b, c). In other cases, however, low similarity was simply due to partial imitation, e.g., in the lineage (LB12), where the song imitation became worse because a grand–pupil dropped a syllable during imitation (Fig. 5d). These findings suggest that grand-pupils of impoverished-song grand tutors imitate some elements from the deficient songs of their tutors, but they also further "balance" them, thus increasing the diversity of their songs.

**Balanced imitation of vocal state abundances**. Our measures up to now summarize the distribution of vocal states within a song. We next looked at each vocal state separately and measured how frequencies (abundances) of vocal states are imitated. In prior studies, we noted that vocal imitation in zebra finches is inversely related to model abundance. That is, too much exposure to a tutored song could inhibit learning[31]. Here we test if this is the case also for abundances of vocal states within a song.

We partitioned the vocal state data into quartiles based on the overall acoustic similarity between tutor and pupil songs. For each tutor–pupil pair, in each quartile, we then plotted the relative abundances of all 10 corresponding vocal states in the tutor's song versus his pupil's song (Fig. 6a–d). We found that relative abundances of all 10 states were correlated, for each quartile. As expected, tutor–pupil vocal state abundances were more strongly associated when imitations were accurate; for example, the residual coefficient of determination was much higher in the top similarity quartile, explaining about 35% of the variance in cases of highest song similarity (Fig. 6a), and only about 9% of the variance in the bottom quartile (Fig. 6d). We noted that in all quartiles, the slope of the correlation was less than one (Fig. 6a–d), meaning that when tutor's vocal state was low in abundance, pupil's vocal states tended to be higher (above the diagonal) and vice versa.

We next tested for statistical significance of this bias across the entire data set. Our null hypothesis is that when the abundance of a vocal state in the tutor's song is high, his pupil is not more likely than chance to deviate from the model in a manner that "balances" his song. In other words, if deviations (imitation "errors") are random, then the likelihood of deviations (errors) to increase or decrease song diversity should be determined by the overall distribution of errors in our sample. In a previous study[2], some of us presented evidence that imitation of isolated tutors is biased: syllables with high abundance in abnormal isolate tutor

**Fig. 5 Song diversity across generations. a** Song similarity across two generations of pupils (colors represent quartiles (as in Fig. 1a)) in 14 family lineages. **b** An example from lineage HP10 showing a transition from poor imitation in a first-generation pupil to accurate imitation in a grand pupil. **c** Same as **b** for lineage DG4. **d** A counter example in lineage LB12, where the grand pupil imitated poorly. Source data for this figure is in Supplementary Data File 1.

song (>20%) were often less abundant is pupil's songs. Using the same 20% threshold we found that the distribution of tutor vs. pupil vocal state abundances is asymmetric (Fig. 6e): when tutor's vocal state abundance is above 20%, about 14% of corresponding pupil's states are above the diagonal (hence 86% of the errors increase song diversity). But looking in reverse, we found that when a pupil's vocal state is above 20%, a higher proportion of corresponding tutor's states (23%) are to the right of the diagonal. To overcome dependencies between vocal states, we treated each tutor–pupil pair as a statistic. We randomly shuffled the direction tutor->pupil vs. pupil->tutor (without breaking the pairs) to obtain a random distribution of biases. We found that the observed bias to increase song diversity (namely in the direction that decreases the abundance of vocal states that are already of

high abundance) is higher than expected by chance (bootstrap direct $p$-value = 0.032).

We wondered if this bias is stronger in cases of poor imitation, due to the inclusion of non-tutor syllables (via improvisation or innate vocalization). To evaluate if this was the case, we divided the tutors' vocal states into 0.1 abundance bins, then calculated the median abundance of pupil vocal states for each bin. For each bin, we calculated the abundance ratio for that median. For example, if at the window centered at 0.1 tutor abundance, the median pupil vocal state abundance was 0.2, then the gain ratio would be 2. A gain value of 1 ($y$ axis in Fig. 6f) represents the identical abundance of all 10 vocal states in pupil and tutor. A gain value of 2 indicates a doubling of abundances in the pupil (amplification), and a value of 0.5 halving (attenuation).

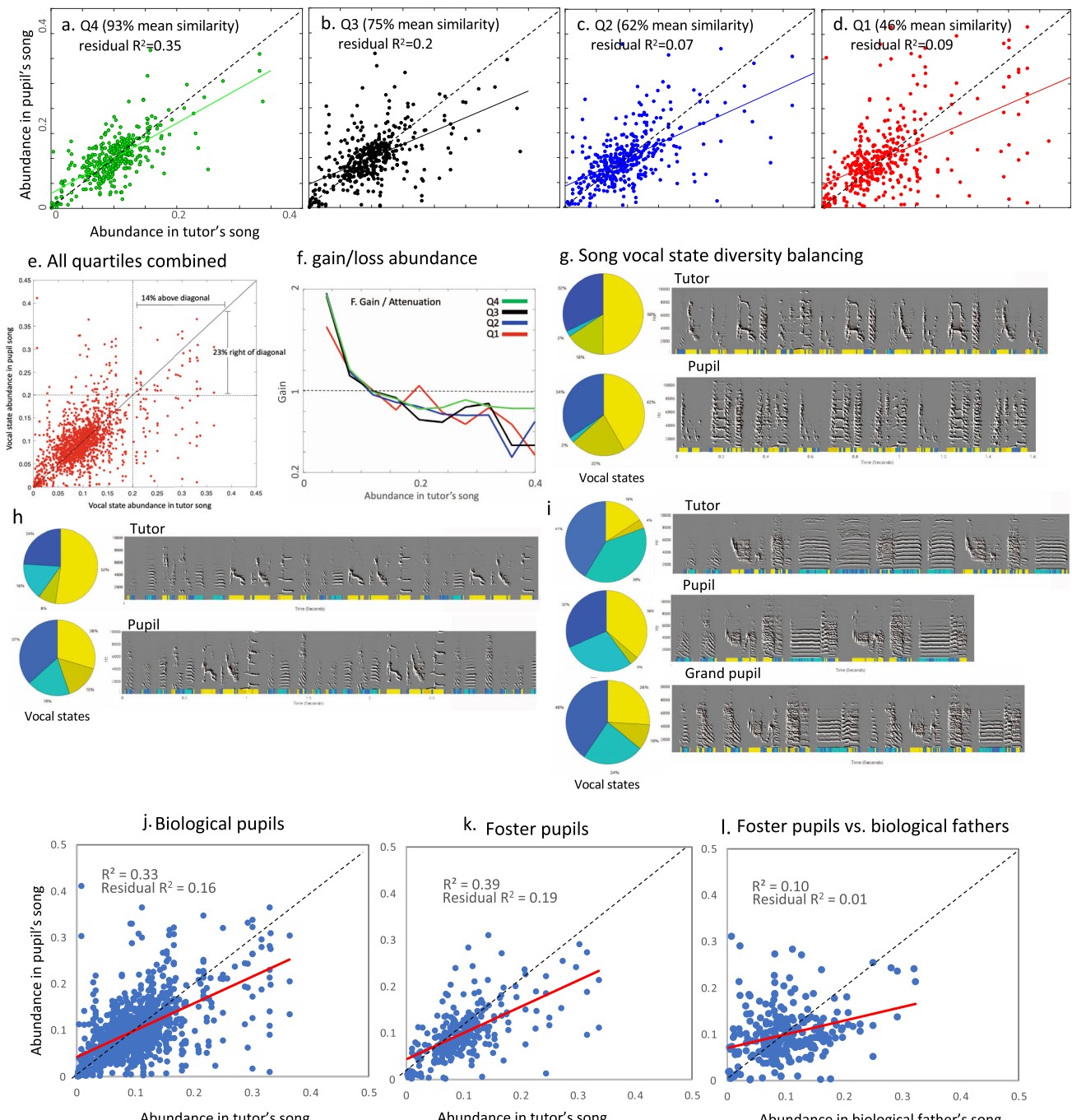

**Fig. 6 Imitation of vocal state abundances. a–d** Scatter plots of tutor vs. corresponding pupil vocal state abundances according to quartiles of song similarities. Note that each bird is represented by 10 markers, which are not statistically independent. The residual correlations were computed after removing trends with bird identities included as random factors. Dashed lines are identity, slope = 1. Colored lines are regression of the data. **e** Same data as in **a–d** combined, comparing vocal states abundances > 20% in tutor vs. pupil songs. **f** Median imitation gains for all state abundances, according to imitation quartiles. Gain of 1 indicates no bias, gain of 2 indicates doubling of abundance, and gain of 0.5 halving. *Y* axis is log-scale. **g–i** Examples of song diversity balancing. We simplified the 10 vocal states into 4 groups: yellow for high pitch states 9–10; mustard for medium pitch, high entropy states 6 and 8; light blue for non-modulated states 4 and 7; and dark blue for the rest 1, 2, 3, and 5. In **i**, we present two generations of pupils. Note the more uniform pie charts in pupils compared to their tutors. **j**, Vocal state abundances in biological tutors vs. pupils' songs. **k** Vocal state abundances in foster tutors vs. pupils' songs. **l** Vocal state abundances in fostered pupils vs. their biological fathers, who did not raise them, which did not raise them. Dashed lines are identity, slope = 1. Red lines are regression of the data. Source data for this figure is in Supplementary Data File 1.

Interestingly, the gain-loss curves have similar shapes and magnitude across all four quartile groups (Fig. 6f). In all cases, a gain of 1 (where abundance tends to be identical across pupils and their tutors), was at 11–12% abundance, which is fairly close to the center of the distribution (=10%, since we have 10 vocal

states). These findings suggest that the regression we noted is not an entirely random effect. For example, in Q1, where the mean similarity is 93%, we see that when tutor state abundance is above 0.2, the corresponding pupil abundance is lower in 10 out of 11 cases (Fig. 6a). In all these cases, the corresponding vocal sounds

were imitated, but produced either less often, or with biased features, by the pupil.

To visually compare vocal state abundances in tutor vs pupil songs, we reduced the ten vocal states into four color codes, and graphed them along with the sonograms of each bird (Fig. 6g–i). In cases where the tutors' songs included many high-pitched vocalizations (vocal states 9 and 10), their pupils imitated, but lowered the pitch, thereby decreasing the abundance of those states (Fig. 6g, h). In another example, where the tutor's song had a high abundance of harmonic stacks (states 4 and 7), their pupil imitated only a subset of these sounds (Fig. 6i). In turn, in the following generation, the pupil's pupil further differentiated his song to include more balanced vocal states (Fig. 6i). Taken together, song imitation appears to be highly sensitive to the relative abundances of vocal states, suggesting a balancing mechanism that prevents song diversity from becoming too low, perhaps independently of imitation.

Finally, we asked whether fostered pupils imitate their tutor's song vocal states as accurately as biological pupils. Analysis at the level of vocal states allowed us to compare how abundances of vocal states are influenced by foster vs. biological fathers. For reference, imitation of vocal state abundances between the 130 biological pupils and their fathers had a residual $R^2 = 0.16$ (Fig. 6j; $t = 5.9$, $p = 3.9e-09$). The 30 foster pupils relative to their foster fathers had a similar $R^2 = 0.19$ (Fig. 6k; $t = 2.5$, $p = 0.01$). This is supported by a near-zero correlation between fostered pupils and their biological fathers (Fig. 6l, residual $R^2 = 0.01$, $t = 0.46$, NS). Therefore, the similarities we observed in vocal state abundances between tutors and their pupils reflect learning with no detectable genetic effect at this level of analysis.

**How balanced imitation constrains distributions of song features.** We first tested if abundances of specific vocal states are similar across low-diversity and high-diversity tutor songs. We pooled together songs from tutors that had the lowest diversity (bottom quartile) and calculated the diversity of their "pooled song". We found that the diversity increased from a mean of 2.99 bits to 3.17 bits, which is similar to the mean diversity in the top quartile (mean = 3.16 bits) but lower than the pooled diversity of the top quartile (=3.27 bits). This outcome indicates that the distribution of vocal states pooled across low-diversity songs is fairly broad, but not as broad as that across songs of high diversity. The distribution of abundances of pooled vocal states (Fig. 7a) explained this difference: As opposed to the nearly flat distribution of vocal state abundances in the high-diversity songs, low-diversity songs tend to have a higher proportion of states 9 and 10, which correspond to high pitch sounds. This is interesting because, in this respect, the low-diversity songs are structurally similar to isolate songs, which are often of higher pitch[32]. As expected, comparing top and bottom quartiles of influence on the pupil show a similar outcome (Fig. 7b). This outcome suggests that mean song features of low and high influence songs should differ. Further, the variance should also differ: High-diversity songs by definition cannot be extreme in their mean feature values. Low-diversity songs can, in principle, have average features that are close to the population mean, but are more likely to have extreme mean feature values. For example, a song containing mostly high-pitched sounds is both low diversity and extreme in its mean pitch (see for example tutor HP10 in Fig. 5b).

We asked whether we can predict imitation outcomes based on the mean features of a tutor song. If songs of low diversity were culturally transmitted less than high-diversity songs, then songs with extreme mean features—which are typically of low diversity —should be transmitted less. To evaluate this, we plotted the mean pitch of tutor songs against the pitch of their pupil's songs.

Indeed, the distribution of mean song pitch was tighter for the top quartile of tutor–pupil song imitation (Fig. 7c). For example, all tutor songs with a mean pitch above 2000 Hz were of low influence (Fig. 7c, histogram red symbols); these extreme songs were also of low diversity. A similar effect can be seen in Wiener entropy (Fig. 7d) and frequency modulation (Fig. 7e): in both cases the distributions were broader for low-diversity songs. Further, for mean pitch, top influence (green line) is equal or higher than low influence (red line) between 795 Hz and 1885 Hz (Fig. 7c). Bottom influence is higher between 1885 and 3000 Hz (red line above green line, Fig. 7c).

We superimposed these empirically determined pitch intervals (for top and bottom influence) on ranges of mean song pitches obtained in a database of four zebra finch colonies including the current one, and shaded the intervals values green (presumably top influence) and red (presumably low influence; Fig. 7f). We then did the same for frequency modulation (Fig. 7g), and Weiner entropy (Fig. 7h). Across the colonies, the distribution of mean song features was to a large extent confined within the range of high influence in our colony. Therefore, the range of mean feature values of highest imitation influences in our colony, but not of lowest influences, seems consistent across zebra finch colonies. This range, in turn, can be explained by balanced imitation as high influences are associated with high tutor song diversity. In sum, this outcome is consistent with the notion that over generations, songs of high feature diversity are more influential, and therefore shape the overall distribution of mean song features in a similar manner across colonies.

## Discussion

We analyzed song learning statistics in a large zebra finch colony. We observed high variability in song imitation outcomes across families. It did not stem from genetic variability, but rather was explained by an environmental effect: the acoustic structure of the tutor's song. Tutors who produced songs of high acoustic diversity had greater influence over the songs of their pupils. In order to more thoroughly study this relationship, it was necessary to develop a measure of song diversity at the sub-syllabic level, where we detected 10 vocal states common across zebra finch songs. We found that pupils copy the abundances of vocal states in their songs from the songs of their tutors, but that they do so in a balanced manner, such that highly abundant vocal states in a tutor's song become less abundant in their pupil's song and vice versa. We discovered that extreme mean song features, which are associated with low song diversity, are also associated with poor imitation. The converse is associated with good imitation. Similar moderate mean song features were more highly present in three independent colonies of zebra finches, suggesting a species-specific mechanism that can be explained by an innate bias to produce acoustically balanced songs. Our findings suggest that this bias is highly sensitive to even mild fluctuations in vocal state abundance in the tutor's song and is independent of imitation outcomes. We call the process "balanced imitation". We suggest that balanced imitation prevents vocal cultural learning from converging too much into complete uniformity or diverging too much into chaos[23]. In such extreme cases, the communication system might become deficient: high song uniformity could reduce individual identity signal, whereas a chaotic song culture might weaken group identity. Whether balanced imitation is ecologically adaptive in zebra finches remains an open question.

Given that the mean song diversity of birdsongs in our colony was close to the upper theoretical limit of diversity, one might wonder why tutors with low-diversity songs are not rarer in our colony. We observed that low-diversity songs are often corrected toward high-diversity songs in the grand-pupils of low-diversity

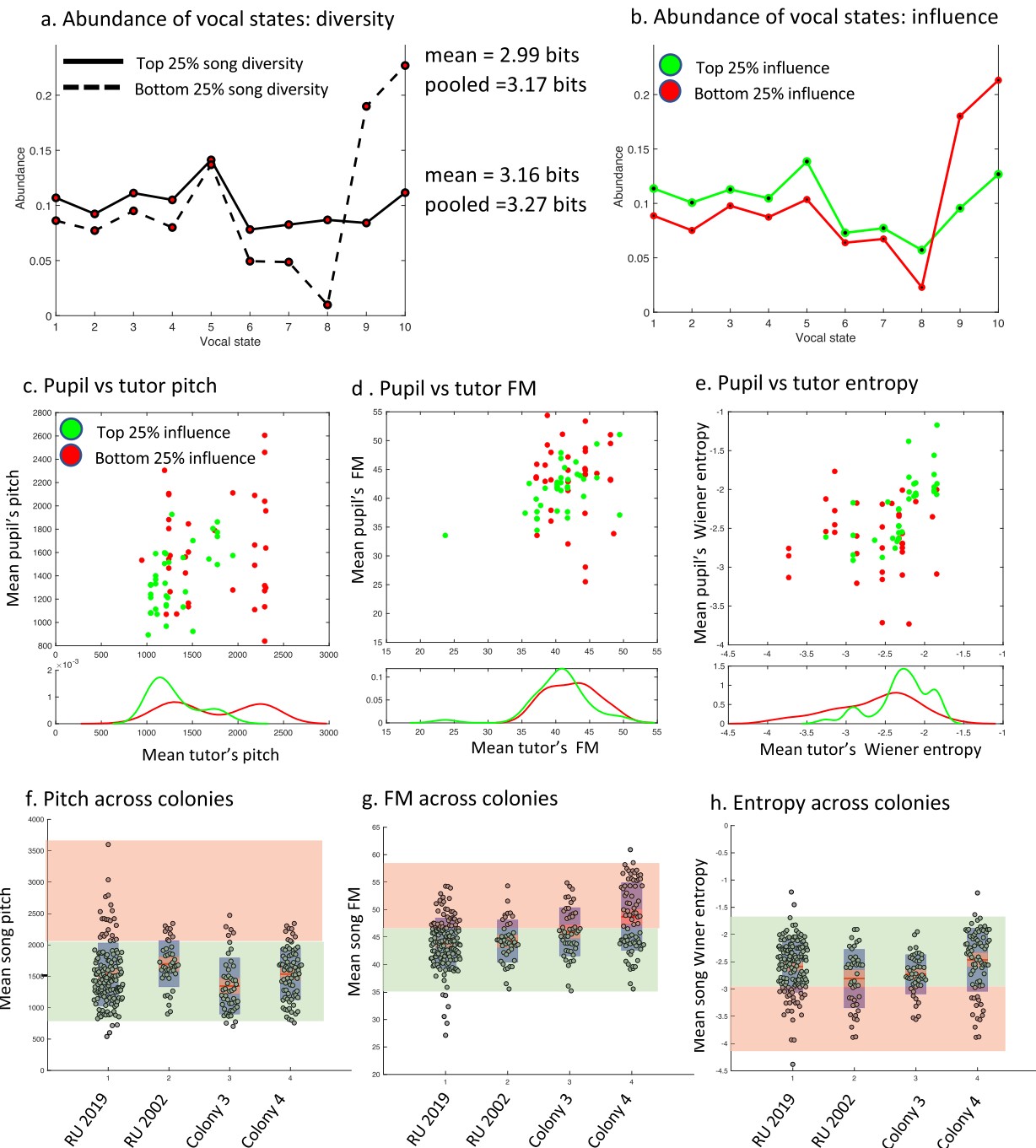

**Fig. 7 Song diversity versus imitation. a** Vocal state abundances in pupils pooled over birds with lowest (bottom quartile) song diversity (dotted line) vs. top quartile (solid line). **b** Same as **a** for the bottom (red) and top (green) quartiles of tutor song influence. **c–e** Mean tutor's song features versus pupil's song features for pitch (**c**), frequency modulation (**d**), and Wiener entropy (**e**) for the top influences (green dots, top quartile) and for bottom influences (red dots, bottom quartile). Plotted at the bottom are histogram lines of tutor features for top and bottom quartiles. **f–h** Box plot distribution of mean song features in four colonies for pitch (**f**), frequency modulation (**g**), and Wiener entropy (**h**). Each marker represents the mean value for one bird. Green shaded areas correspond to top influence feature ranges in colony RU 2019 (this study), whereas red shaded areas correspond to bottom influence feature ranges in colony RU 2019 (n = 149 birds). In the box plots themselves, the red line is the median; Orange fill are the upper and lower quartiles; Blue fill is the minima and maxima. About 20% of the RU 2019 colony are descendants from the RU 2002 colony (Rockefeller Nottebohm Lab; n = 42 birds). The remainder of the 2019 colony originated from Duke University. Colony 3 is from the University of Southern California (Bottjer Lab; n = 48) and Colony 4 is from Cornell University (Regan Lab; n = 77). Source data for this figure is in Supplementary Data File 1.

song tutors. This correction via cross-generational dynamics may sustain homeostasis of high song diversity. But it is not unlikely that other social forces drive songs toward low diversity. This study, as well as previous ones, reported strong variability in song imitation success across birds. For example, we previously

reported social inhibition of song learning in clutches that contain more than two male siblings[33]. Such partial imitation of the tutor's song due to social inhibition may lead to impoverished songs. The current study confirms that a proportion of pupils of high-diversity tutors acquired very low-diversity songs, perhaps

due to social inhibition of song imitation, or due to lower genetic capacity for imitation. Finally, we note anecdotally that some of these poor song learners were aggressive, dominant birds that were among the best breeders in our colony, in terms of the number of offspring sired. This is counterintuitive, given that zebra finch females are less likely to ovulate in response to males who imitated poorly[33]. Further studies should look into the fitness consequences of such phenotypes.

The current study generalizes upon the previous studies[17,34,35] that documented the emergence of song culture from the impoverished song of isolated founders[2,15]. Songs of isolated zebra finches often contain an abnormally high abundance of some song elements. Pupils of isolate tutors normalized these high abundances in their imitations, reducing them to the species typical range. Our statistical analysis suggests that in those studies[2], song normalization across generations was driven, at least in part, by the balanced imitation discovered here: song syllables that were abnormally high in abundance in the isolate tutors were copied at lower frequencies, and vice versa, syllable types that were rare in the isolate song were copied at a higher frequency. Interestingly, evidence suggests that this effect is not driven by any overall abnormality of the songs. Instead, it occurs when tutor songs are well within the wild-type species typical range, at the micro level of balancing vocal state abundances. Further, biases in the statistical learning of vocal state abundance can explain variability in imitation outcome across families, and allow us to distinguish between low song similarity due to putative imitation failure, as opposed to "corrective" deviations, leading to a cross-generational homeostasis of songs with high acoustic diversity.

It would be interesting to test if balanced imitation parameters are different across species. Variation in the intensity of the trend to sustain high song diversity and that of the trend to imitate songs accurately could lead to equilibriums that differ according to species and possibly even the ecological conditions in which a species lives. Perhaps species with songs that are similar across individuals engage in weak balanced imitation and vice versa. For example, to explain why the songs of the Timor zebra finch are much more similar across individuals compared to the Australian zebra finch[25], we speculate that perhaps a weaker balanced imitation gain in the Timor zebra finches (compared to Australian zebra finches) could potentially increase the odds of extinction of rare song elements, driving the stronger convergence observed in songs across individuals.

Regardless of possible prevalence across species, accounting for balanced imitation in zebra finches might be necessary in order to properly interpret vocal learning outcomes. This is particularly important because mechanisms of vocal learning are studied extensively in Estrildid finches, among which song learning outcomes vary considerably across individuals. In part, this variability is associated with factors like genetics and with tutoring mismatches[12,36]. Our results indicate that, in addition, deviations from tutor song through reorganization and transformation of copied vocal sounds may be driven by an inclination to optimize song diversity. This can be regarded as a discrete form of error correction during song learning. That is, balanced imitation involves correcting errors from states of minimal (and perhaps also maximal[34]) diversity. In the framework of error correction[37], the developmental question is when and how the vocal learning bird balances between error correction exclusively in reference to tutor sound to error correction in reference to a state of its own sound diversity. A better understanding of this balance and possible transition could reveal the mechanism through which a species-specific level of cultural song diversity is determined[23].

Another observation that requires further study is the recombination of syllable units. In the cases we observed, pupils combined tutored syllables into new and more complex units, but splitting appears to be rare. Splitting could be an artifact due to limitations of our methods in detecting such recombinations; it could also suggest a tendency to compress the tutored song. Such a compression might be useful when several potential tutors are available. Compressing the imitation from one song could leave more room to imitate song elements from other tutors. Further, some improvised syllable types tend to be acoustically simple and are transformed across generations into complex types[2]. This line of thinking suggests that perhaps we should consider not only the overall acoustic diversity of a tutored song, but also the diversity per unit time. Here too, variation across species can be potentially explained: as opposed to zebra finches, in Bengalese finches, syllable level analysis shows correlation in song and transition diversity across tutors and pupils[38]. We do not know if syllable recombination is common in Bengalese finches, but they usually produce less complex syllable types compared to zebra finches, which could suggest that Bengalese finches are less inclined to compress their songs.

Previous studies[25,39] suggest that high song diversity in a colony of zebra finches could be adaptive. In the zebra finch female, brain dopamine response to songs is tuned to the song of her mate[39]. To the extent that balanced imitation can also sustain the acoustic diversity of songs within a colony, it might also enable the females to respond selectively to their mates. Balanced imitation is also of interest in a broader context of vocal and non-vocal cultures in humans. In general, cultures may vary in their stability and in their richness (polymorphism), and balanced imitation could potentially explain how different morphs of cultures come about. At the population level, balanced imitation can be thought of as an example of a balancing (negative frequency-dependent) selection of morphs, which can promote polymorphism by preventing the extinction of rare morphs. At the individual level, it can be thought of as a mechanism that promotes diversity in the skills that are acquired. It would be particularly interesting to test how imitation biases might interact with the structure (topology) of communication networks, in determining how cultural behavior spreads and is filtered over space and time[40]. Finally, other possible mechanisms could potentially explain balanced imitation including perceptual biases[41] and habituation[42].

In conclusion, by recording and analyzing families of vocal learning birds in a large colony, we have gained a deeper understanding of mechanisms that constrain the learned vocal repertoires of a species. This mechanism may regulate the level of convergence or divergence in long time scales across generations, while sustaining a certain level of acoustic diversity within a population. While our study is far from exhaustive, publishing our song imitation library at the Linguistic Data Consortium catalog (https://www.ldc.upenn.edu) and in http://ofer.hunter.cuny.edu/songs should allow others to further test the generality of our findings using different approaches at different degrees of granularity.

## Methods

**Animals**. All experiments were approved by the Rockefeller University IACUC. Each bird was raised by his parents in a flight cage until day ~90. Partitions kept each family visually isolated from neighboring flight cages. Cages were distributed across three rooms (~250–500 sq. ft each).

**Audio recording**. Birds were placed singly in sound attenuation chambers[43] and their vocal activity was recorded continuously over one week, in cohorts includes 8 zebra finch males in 8 boxes (Box 1, Box 2,…, Box 8), recorded simultaneously over a week, using Sound Analysis Pro[28]. By definition, these are undirected songs. All songs analyzed in this study are undirected songs. All song recording data were generated at the Rockefeller University Field Center Colony between July 2018 and Aug 2019.

**RU 2019 zebra finch song library**. All song recording data were generated at the Rockefeller University Field Center Colony. The library currently includes recordings from 280 birds, including 160 tutor–pupil pairs. In addition to recording undirected songs, most birds were also recorded singing female-directed and male-directed songs.

**Songs libraries from other zebra finch colonies**. We used the zebra finch song data set published at http://people.bu.edu/timothyg/song_website/index.html, which includes recordings from individual zebra finches from different colonies. We used data from three colonies: Nottebohm lab (RU 2002; The Rockefeller University), Bottjer lab (Colony 3; University of Southern California), and Regan lab (Colony 4: Cornell University). The songs were downloaded, and mean features: pitch, frequency modulation, and Wiener entropy were extracted using Sound Analysis Pro. One of the colonies in the database, RU 2002, is related to our colony: In 2017 approximately 150 birds that were descendants of the RU 2002 Millbrook colony were mixed with approximately 300 birds brought from Duke University. Recordings are also 15 years apart, past the lifetime of a zebra finch, meaning that none of the RU 2002 birds are the same as the RU 2019 birds.

**Data analysis and statistics**. All data were analyzed using MATLAB and R, including the Sound Analysis Tool package for MATLAB, which was used to extract song features. We deposited our MATLAB code for computing song diversity together with the entire song-library raw data at the Linguistic Data Consortium (URL and access # pending).

**Similarity measurement**. For each tutor–pupil pair, we calculated similarity measurements with Sound Analysis Pro 2011[28] using the default settings. Briefly: the similarity estimate is based on four-song features: pitch, frequency modulation, Wiener entropy, and spectral continuity. The similarity matrix is then computed over 70 ms windows of these feature vectors, followed by detection of continuous similarity sections. The overall similarity estimate is the proportion of a tutor song motif that is included within similarity sections with $p < 0.05$. For each song, we outlined a motif and calculated %similarity of tutor vs. pupil song (asymmetric song similarity). We repeated this calculation five times and used the median % similarity as our estimate for each tutor–pupil pair.

**Analysis of syllable-type diversity**. For each bird, we sampled song bouts containing at least 1000 syllables per bird, which we then automatically segmented and clustered using Sound Analysis Pro. After a semi-automatic segmentation to syllables based on amplitude threshed, we calculated mean syllable features: mean pitch, mean frequency modulation, mean Wiener entropy, and mean Spectral continuity, to summarize the acoustic structure of each syllable. We then performed a hierarchical nearest neighbor clustering to identify syllable types on the entire sample. This method identified clusters (syllable types) and automatically counts how many syllables appear in each cluster. The relative frequencies were calculated from these counts.

**Evaluation of syllable-type recombination**. Our measure of syllable-type diversity is based on classifying syllable types within each bird; it does not evaluate or compare syllable types across birds. For evaluating syllable-type recombination, however, we need to determine which syllables were copied, either as a unit or in parts. Using similarity measurement with Sound Analysis Pro, for each syllable in tutor song we automatically detected sections of similarity in pupil's song. Evaluating these sections allowed us to determine if the boundaries are consistent across tutor's and pupil's songs. In particular, we could detect splitting (a similarity section with one interval in tutor song and two disjoint intervals in pupil's song) or merging (a similarity section with one interval in pupil song and two disjoint intervals in tutor's song). Note, however, that we only included cases where imitation was subjectively apparent, which is easier to determine in cases of merging compared to splitting. Therefore, our lower bound estimates could be biased toward the detection of merging.

**Classification of vocal states**. For each bird, we analyzed singing bouts of 6–8 s each. We used the Sound Analysis for Matlab tool box (http://soundanalysispro.com/matlab-sat) to calculate song features using the default settings of FFT window size = 10 ms in steps of 1 ms. We set a 50 dB threshold (uncalibrated, baseline = 70 dB), below which data were regarded as silences and excluded from further analysis. For each 1 ms window, we calculated vocal states as follows: We first detected the pitch category (see boundaries in Fig. 4a). We then identified clusters in each slice according to the boundaries of the heatmaps outlined in Fig. 4b. We chose to use this simple method in order to cover the entire vocal space of the bird. That is, we classified each FFT window as belonging to one of the ten states, without residuals.

**Calculation of song diversity**. For each bird's song (including bouts of 6–8 s), we calculated vocal states as shown above. We then calculated the proportion of vocal sounds within each vocal state and calculated information entropy[27] over the 10 clusters, $-\sum p_i(\log_2(p_i))$, where p = the proportion of sounds within each vocal state $i$.

**Fixed effects statistical models**. For statistically independent measures such as syllable diversity, vocal state (song diversity), similarity and influence, we used the Matlab fixedEffects(lme) function. Where lme is a statistical model such as "PupilSongDiversity ~ TutorSongDiversity".

**Linear mixed-effect statistical models for vocal state abundance**. We used the Matlab linear mixed-effects model fitlme function for all statistical analysis unless stated otherwise. All statistical models and results are presented in the supplementary information. For statistically independent measures (one measure per bird) such as syllable diversity, vocal state (song diversity), similarity and influence, we still have dependency due to tutors raising several pupils and we, therefore, consider tutor and pupil identities as random effects. Because abundances of vocal states are not independent even at the bird level, with some vocal states being more frequent than others in the population, we first need to account for this global trend. We refer to $R^2$s as "residual" to indicate the removal of this global trend. Prior to statistical analysis, we removed the overall trends in vocal state abundances, so as to make each vocal state equally abundant in the population, which guarantees a zero correlation when tutor and pupil identities are shuffled (which we confirmed by shuffling). Because of repeated observations (10 states per tutor–pupil pair), we accounted for random effects of both tutor and pupil identities using mixed-effect models. See Supplementary Note 2 for a complete description of the models.

**Statistical treatment of multiple tests**. p-values that we call "statistically significant" are all $< 0.01$ after Bonferroni adjustment.

**Reporting summary**. Further information on research design is available in the Nature Research Reporting Summary linked to this article.

## Data availability
All raw data have been deposited in a Github: https://github.com/oferon/Balanced-Imitation. Further, our entire zebra finch song library (.wav files), family trees of all individuals, example sonograms, bird IDs, and more have been deposited in a public Dropbox folder: https://www.dropbox.com/sh/vvrz3o2inb1ynxk/AAAI6oaJ_pkrML8ON_kQa-_xa?dl=0. We also created a web portal to the data: http://ofer.hunter.cuny.edu/songs/the-rockefeller-university-field-research-center-song-library. Source data are provided with this paper.

## Code availability
We deposited all Matlab scripts in a Github: https://github.com/oferon/Balanced-Imitation.

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

## Acknowledgements

We thank Michael Yartsev, Lucas C Parra, and Lomax Boyd for useful comments and suggestions, and the Rockefeller University Field Research Center staff Gillian Permuy and Lotem Tchernichovski for helping with audio recordings. Supported by NIH grant NIH DC04722 to O.T. and by HHMI and Rockefeller University funding to E.D.J.

## Author contributions

O.T. designed the study, collected data, performed analyses, and co-wrote the manuscript; S.E.-E. performed analyses and co-wrote the manuscript; E.D.J. helped design and coordinate the study, and co-wrote the manuscript.

## Competing interests

The authors declare no competing interests
