## [Peer Review File · Nature Communications]

REVIEWER COMMENTS

Reviewer #1 (Remarks to the Author):

In this study, the authors track the cultural transmission of songs within a large multi-generational colony of zebra finches, and identify a previously unknown factor that shapes vocal culture. Zebra finches learn to sing as juveniles by imitating the songs of their (genetic or foster) fathers. It has been long known that learning outcomes are very variable, and this variability has been accounted for by variability in learning capacity and by innate species-specific biases (namely, pupils “correct” abnormal tutors towards a more species-typical song). However, here the authors show that independently of these two factors, imitation variability can be explained by pupils’ tendency to adjust the diversity of their tutor’s song, such that common elements become less frequent, and rare elements become more abundant. In a similar manner to balancing selection in evolution which maintains genotypic variability in populations, the authors conclude that a frequency-dependent imitation bias in zebra finches maintains a “steady state” of relatively high song diversity through learning generations, and could explain the fact that zebra finch populations don’t converge to local dialects. These findings are very interesting, and are likely to open a fruitful new direction in cultural evolution studies.

The study is well conducted, and the analysis methods are adequate and innovative. In particular, the authors extract a relatively small number of distinct short-time-scale (10 ms) acoustic “building blocks” that characterize the songs repertoire of birds in their colony (and likely, of zebra finch songs in general), and use these to quantify the information entropy (diversity) of songs. This is important, since there is growing evidence that these shorter time scales might be more important to the birds than time scales that are more salient to the human ear, like syllables and motifs. In addition, the data of song samples in identified tutor-pupil families within a colony are made available in a public song library at the Linguistic Data Consortium catalog at UPenn, which is hopefully the first step in building a large cross-study library of song cultures.

All the above make the manuscript a strong contribution that would be of interest to the broad readership of Nature Communications. However, I have several comments regarding the need to supply information about the diversity of tutor songs included in the study, to discuss alternative interpretations of the results, and to clarify some of the reasoning and methodology, as detailed below:

General comments:

1. It would be good if the authors could provide some information about the overall diversity of tutor songs within their sample. Since at least some tutors are ex pupils of other tutors, it would be good to rule out the possibility that low-diversity or high-diversity tutors share some other common features besides acoustic diversity that may influence imitation. Is there a similarity in song features of tutors that were not well imitated (and tutors who were) aside from the diversity?
2. The authors present the balancing selection employed by pupils with regard to their tutors’ song as an adaptation, that “sustains cross-generational homeostasis that prevents the collapse of vocal culture

into either complete uniformity or chaos.” However, 1. It is not clear at this point whether this imitation phenotype is indeed adaptive, and the authors should consider alternative possibilities, such as that it is a by-product or side-effect of the birds’ perceptual biases. 2. The adaptive value the authors attribute to balanced imitation is not clearly explained: why is cross-generational homeostasis important? What is the problem with a vocal culture “collapsing into chaos”, and what is meant by “chaos”? 3. It seems that there is something special about zebra finches compared to other species that have song dialects – how do the authors account for it given their results? Namely, if balanced imitation is what prevents zebra finch vocal culture collapsing into total uniformity, how could species that have song dialects avoid this fate?

3. Overall, the writing is clear and succinct, but in some parts I struggled to follow the authors’ reasoning, and some of the method descriptions were too succinct (at least for me) and need to be more comprehensively explained. These are detailed in the specific comments below.

Specific comments:

Some parts of the abstract could be made clearer, as follows:

Abstract line 18-19: even in cases of accurate imitation, pupils often recombined imitated syllables to form new units, and, therefore, distributions of syllable types in pupils’ songs were not well correlated with their tutors’; then why are these considered cases of accurate imitation? It’s not clear how this explains good song imitation in some families and bad imitation in others.

Abstract: lines 20-23: ... rare vocalizations in tutors’ songs became more abundant in their pupils’ songs, and vice versa. Consequently, cultural transmission of tutor songs that were high in acoustic diversity were stronger than those that were low in diversity. This second sentence seems to contradict the “vice versa” in the previous sentence – if pupils always balanced out tutor’s diversity, why was cultural transmission stronger in higher diversity tutors?

Last sentence of abstract: “... that prevents the collapse of vocal culture into either complete uniformity or chaos.” Very dramatic, but vague. Sounds like something terrible is going to happen if there is no balanced imitation, but what exactly? (see general comment #2#)

Introduction:

Page 2 lines 13: at this point it is not clear what exactly is meant by acoustic diversity: is it the variability across tutor song renditions, or the range of acoustic states within a single song, or something else? This makes the remainder of the paragraph a bit cryptic. This of course becomes clearer after on, but since acoustic diversity is such a central measure in the study, it could help if the authors provide a brief explanation when the term is first mentioned.

Results:

Lines 26-27 (page 2): how did the authors determine there was no imitation across families? This is not explained in the methods

Supp Fig 1: what is the y axis?

Fig 1a: does every “pupil” ellipse represent a single bird or a single clutch of a given tutor?

Page 3, lines 3-6: “In both kinds of cases, the birds still produced their syllables in the typical repeated song motif of 2-6 syllable types (Fig. 1b,c). This suggested to us that pupils might more accurately imitate tutor songs that are rich in acoustic structure (i.e., acoustically diverse), while improvising upon

impoverished tutor songs.” I don’t understand the transition between the first and second sentence – how does the fact that pupils sang repetitive motifs suggest that they imitate tutors with rich songs more accurately?

Page 3, lines 6-7: “To test this hypothesis quantitatively, we compared the diversity of song syllable types in tutors and their pupils.” It would help if the authors specify here the predictions of their hypothesis regarding the correlation between tutor and pupil diversity. Later in the text, the authors express surprise at not finding a positive correlation between tutor and pupil syllable diversity, but why would they expect to find a positive correlation (see point on that below)? It rather seems that their hypothesis suggests there should be an interaction between diversity and imitation accuracy.

Page 3, line 7 – up until now, the authors use the terms simple, complex rich in acoustic structure, and diversity without explaining what they mean.

Page 4, lines 6-7: 1. how was syllable relative frequency (abundance) calculated? Was is the proportion of performing a certain syllable type out of all the syllables in a single song bout? Song motif? Entire song sample of a bird? 2. How did the authors determine that two different birds share a syllable?

Page 4, lines 8-10. The study’s findings center on measuring song diversity using Shannon’s information entropy. Since this measure was not used before to quantify diversity of vocalizations, or motor sequences in general (if it was, the authors don’t cite any), it would be good to provide more detailed information for the reader, and preferably also some means to intuitively assess what is a high entropy or low entropy song.

Supp fig 2: both axes labels are missing.

Page 4, line 11: why is it surprising that there is lack of correlation between tutor and pupil diversity? Not all pupils copied the songs accurately, so why are those expected to be as diverse as their tutors? According to the hypothesis presented by the examples in Fig 1b-c, the authors would expect a that only the diversities of pupils with high-diversity tutors would be correlated with their tutors’ diversity, correct?

Page 4, line 23: rearrangement of tutor syllables should not affect the syllable diversity scores and relative frequencies of syllables, only merging of tutor syllables would.

Page 4, lines 23-25: “We assessed a lower bound estimate of similarity in the syllable boundaries of tutor and pupil songs “. This sentence is cryptic to me - what is meant by lower bound estimate? Did the authors specifically assess syllables that were merged? Do they mean that they restricted their analysis to syllables whose boundaries did not match those of the tutor but that were clearly imitated? The selection criteria for syllable assessment should be more clearly explained.

Fig 3b: 3D plot looks 2D. I don’t see the data on the entropy axis.

Page 6, lines 18-21, Fig 3c legend and methods (page 17, lines 34-36): how were the contours of the pitch distribution, and the cluster boundaries in each pitch slice identified? This is not clearly explained in the methods. Also, the methods refer to Supp Fig 1, but does not show pitch categories.

Fig 3 f legend: all male birds recorded means pupils and tutors, or just tutors?

Fig 3g – what does median indicate, the median of what, diversity in all birds, tutors and pupils?

Page 6, lines 41-42: “However, at least, in the top two quartiles the coefficient of determination seems robust ($R^2 = 0.23$, Fig. 3e) with no apparent interaction with specific family branches.” Isn’t that trivial? If the imitation is good (songs are similar) then the diversity between tutor and pupil should also be similar. In general, I don’t understand the point that panel e is intended to make. It seems trivial that the

correlation between tutor and pupil diversity in bad pupils will be less robust than in good pupils. Is the point of the panel just to show the advantage of looking at diversity across vocal states instead of syllables?

Page 7, lines 9-10: "...such that a large proportion of pupils with low song diversity had tutors with high song diversity, and vice versa." What is meant by a "large" proportion?

Fig 2g: not stated whether the r value is significant.

Page 8, lines 27-28: I don't follow the reasoning: how do the authors know that low similarity was a result of poor imitation in this case, but a result of balancing selection in Fig 4 b and c?

Page 8, lines 41-42: "Assuming a natural trend to develop low diversity songs either via imitation or improvisation,...". "low" = "high"?

Page 8, lines 43-44: "We found that pupils that imitated poorly, regardless of tutor song diversity, tended to have low diversity songs ($R^2=0.20$).". What correlation does the R^2 value refer to?

Page 9, lines 6-7: "Pupils of tutors with high song diversity who imitate well, produce songs of comparably high diversity." Tautological sentence: how can the pupils not produce high diversity songs if they imitate high diversity tutors well?

Page 10, lines 13-14: "to test our hypothesis farther..." which hypothesis is referred to?

Page 10, lines 14-15: why are abundances called raw frequencies? Is this the same abundance calculation that the diversity measure is based on?

Fig 5a: what does raw distribution mean?

Page 12, line 8: Blue lines, do you mean colored lines?

Page 10, lines 34: I don't understand how fig 5e shows regression towards the mean. Would help if the authors could elaborate.

Fig 5e, what does the dashed line indicate?

Page 10, lines 36-37: I don't understand the gain calculation. What exactly is the median calculated over?

Page 13, lines 8-9: "Can we predict imitation outcomes based on the mean features of a tutor song?" I couldn't follow the rationale for asking this question. What hypothesis is tested here?

Page 13, lines 22-25: "Across the three other colonies, the distribution of mean song features was to a large extent confined within the range of good imitations in our colony. Therefore, the range of mean feature values of best imitations, but not of worst imitations, seems consistent across zebra finch colonies." This is unclear to me. How can we distinguish in Fig 6d-f between good and bad imitators in the other 3 colonies? The plots show only the mean feature values for each bird. How can the authors deduce that the birds in the 3 additional colonies that were within the range of good imitation in the first colony were also good imitators?

Last paragraph of the results: I am not absolutely clear about the significance of the finding that balanced imitation reduces the frequency of songs with extreme mean feature values. It is not mentioned in the discussion. Is the point of this analysis to demonstrate balanced imitation across colonies? If that's the case, it might help to change the paragraph's title accordingly.

Discussion:

Page 15, lines 16-17: which of the findings suggest that the imitation bias is very sensitive to fluctuations in abundance?

Page 15, line 20: "...or diverging too much into chaos." See general comment #2 – the adaptive value the

authors impute to balanced imitation needs to be clarified.

Page 15, lines 31-33: "The current study confirms that a proportion of pupils of high diversity tutors acquired very low diversity songs, perhaps due social inhibition of song imitation." But it could also be due to lower capacity for imitation.

Page 15, lines 9: Could the authors speculate on which ecological conditions would favor more or less inter-individual variability (see again general comment #2)?

Minor corrections:

Refs 15 and 19 are duplicated.

Page 2 line 31: ref 27 should be ref 28? Maybe I'm wrong. Does Shannon talk about similarity or just entropy? Same for Page 4 line 5, and other parts of the text.

Reviewer #2 (Remarks to the Author):

Tchernichovski et al. ask how vocal diversity is maintained in a colony of zebra finches. Based on previous work it seems that a colony would converge onto a narrow range of syllables and song complexity. However, this is not the case. The authors analyzed song in a colony to determine how diversity is maintained. They develop methods to analyze 'acoustic diversity' in the colony. They find diversity within a colony is maintained through 'balanced imitation' in which the song of some tutors is well-copied but the acoustic diversity decreases in the pupil, and the tutor songs that are poorly copied increase acoustic diversity.

This is a well-written paper in which the authors take us on a scientific journey of discovery. The ideas are interesting, novel, and noteworthy. The data analysis will be an important contribution to the field and has the potential to be used for by many labs (as other analysis from the OT lab).

I have a few comments for consideration:

1. I was a little confused how the authors define 'diversity'. They discuss acoustic diversity, song diversity, syllable diversity, and vocal state diversity. Are these all the same? In the methods there is a definition of diversity, but it is more statistical, could the authors provide a more intuitive description of 'diversity'?
2. Figure 4. The authors state that there are correlations in Fig. 4e and g. However, it looks like much of the correlation is driven by one or two tutors. That is, if data points less than 2.8 song diversity are removed, it looks like the correlation would be lost. Is that the case? If so, how can those results be interpreted? (Is this a concern?)
3. It would be interesting to have the authors suggest how these ideas could impact other research, outside of songbirds. It seems these ideas are important for other areas of research and it would be fun to propose some additional impacts of this work.
4. One of the findings that is particularly interesting and powerful is the comparison to other colonies. This is part of the final paragraph in the results, but perhaps should be highlighted more.

Very minor points:

5. Line 129. What is Shannon information? Could the authors provide a sentence or two what this

analysis is, and why it is useful? I do not have a strong background in statistics, so it remained mysterious to me. Adding this information would be good for a more general reader.

6. On line 83-84. What is the range of pupils per tutor? This can potentially weight the data and interpretation, as the authors mention later in the paper.

7. Line 258. "all of the grand-pupils imitated more accurately." Than the father, or the grand-father? This is a little unclear as written.

8. Line 259. "the grand-tutor songs were atypical." Atypical compared to what?

9. Line 280. Add reference to Fig. 4g (missing in the text).

10. Line 315. "To test our hypothesis further..." it was not clear what the hypothesis was at this point.

11. Last paragraph of results. It is worth reminding the reader that the data analyzed are from RU2019. It seems that RU2002 is from an earlier time in the same colony, which is interesting to see how the colony changed (or did not change) over time. The author state "Across the three other colonies,..." Should this be two other colonies, since RU2002 is technically the same colony? This was a little confusing.

12. Line 484. "Our statistical analysis suggest" – suggests

13. Line 503. Add 'Australian' to zebra finches.

Reviewer #3 (Remarks to the Author):

Review of "Balanced imitation sustains song culture in zebra finches"; by Ofer Tchernichovski, Sophie Eisenberg-Edidin and Erich D. Jarvis.

This MS presents a detailed analysis of similarities and differences between the songs of tutor and pupil zebra finches. The authors have used a large number of tutor-pupil pairs in the analysis, using multiple metrics and techniques to quantify and compare song learning outcomes. They show convincingly that pupils tend to normalize their songs, including new material when they have relatively poor tutors, here defined as tutors with a small repertoire compared with the norm. In terms of the amount of data, quantification and analysis, the MS is impressive. Its only real weakness in these aspects is a lack of detail on how sounds were measured and, in particular, compared quantitatively to derive measures of similarity that are central to the work. Techniques used are likely valid, but cannot be assessed with the methods presented.

The authors conclude that the pupils tend to 'balance' their songs and name this process "balanced imitation". While this is a valid theory, I have some comments/ concerns. One is that when pupils add new syllables to their songs, this term suggests that they imitate these syllables, but where from? The authors discount early on any learning from neighboring tutors. This assertion however is weak - the birds can hear each other and there is no evidence provided that this does not occur. If the authors are correct though and there is no imitation from neighbors, then presumably the pupils are developing the

syllables de novo with some innate syllable template. If this is the case then "balanced imitation" is a misnomer.

My other concern is that the authors present "balanced imitation" as an active process. What is the evidence though this is not just a random process? That pupils listen to their tutors to some extent and just randomly add things if the tutor's song is not that interesting? If a species maintains a certain level of diversity in its songs (as demonstrated nicely by the authors), then it seems reasonable to assume that there is an underlying innate level of diversity that they use/learn/prefer. Perhaps this tendency to a certain diversity is not random and perhaps it is driven by a desire to have 'balance', but I do not think that the authors have demonstrated this convincingly. They must be able to refute the null (random) hypothesis.

The MS itself is very well written with excellent use of language. It is clear (if, at times, complex) and highly readable. The figures are excellent and provide a wealth of information, but at times become too small to read, and it is likely that parts can be removed. They are very good and thorough, but perhaps overly-thorough.

Abstract

Excellent, v clear and easily understood.

Introduction

Line 53: "more variable across individuals";. Slightly confusing. Each individual has a high variability, or the individuals are different to each other? In other words, does this mean variability within or among individuals?

Results

Line 75: Visually but not acoustically isolated from other families, so song learning across families cannot be ruled out. Found "no evidence of song imitation across families"; But would this not be difficult to detect due to variability?

Line 76: So were the birds removed and placed in isolation for a week or so for recording?

Figure 1a and text lines 84-86: So you used mean clutch similarity rather than individual sibling similarity within the clutch which seems reasonable. But then Fig 1a is a little misleading as it only indicated the individual offspring with no indication of which were from the same clutch. While it's a nice figure, it's displaying information that is perhaps not relevant or, at worst, confusing as the individual was not necessarily the basis for analysis. Additionally there is no way of knowing which of the pupils were clutch mates and so no way of visually assessing how this might affect individual sibling learning.

Line 132-134: Is the Shannon entropy measure used as the measure of syllable type diversity? So in fig 2c, are axes effectively the result of the entropy equation given in line 132? It's not clear. A simple measure of diversity is just mean number of syllable types produced per song which might also make sense as the values of the axes of fig 2c. Please clarify in text and figure.

Line 150-151: You say that you restricted this analysis to where there was full imitation but the boundaries had changed. You make the point that all cases involved merging of syllables rather than splitting. But is this not inevitable if you restricted this to cases where the syllables could be identified? If the syllables were split, they wouldn't be recognizable as the same syllable any more. If this is the case, then the last sentence of this paragraph is a bit misleading as it suggests that splitting could have been seen but was not.

Figure 2: Nice figures, but you don't indicate anywhere up to this point what "FM" and "Weiner entropy" are. You don't mention them in the caption and then, in the text, when you mention them you flick straight on to Fig 3. In addition it's annoying having the y-axis labels for those two data sets in 2b hanging over the data from the previous data set. These graphs should be better separated (or deleted as you're really talking about them in Fig 3).

Lines 173-175: A few things here need changing or explanation. Single FFT windows of only 10ms won't give information on frequency modulation; you need some longer time span to do this, so this doesn't make sense. Also, I hate to be a pedant, but you might be measuring frequency, not pitch? Pitch is the perception of frequency and is logarithmic. Looking at fig 3 the frequency scales on 3c are linear but for 3b are log (although I can barely read it).

Line 176: "all birdsongs"? I suspect you mean all the songs of the birds in your study population. I initially thought you meant all species of song birds.

Lines 183-184: inconsistent use of "&" and "and" between numbers.

Line 198: remove comma after "least";

Figure 3: although it's packed with useful information, the sub-figures are too small to be able to see details let alone read axes. X-axis of 3d in particular needs something. Suspect the dotted line is 0 with negative to the left and positive to the right?? 3b looks potentially interesting but can't see, on the face of it, clear reasons for the partitioning used. Might be better just to keep 3c and d? Also I'm not sure that f adds much as it's essentially the same data as in e and g - the long left tail.

Lines 204-207: Is this becoming circular? You used the existing songs to define the 10 vocal states, and then are pleasantly surprised that the median entropy is close to the maximum, i.e. surprised that there are lots of sounds in each of the vocal states that you used the sounds themselves to define. Why don't you have 12 states with 0 FM states between 1 and 2 and between 9 and 10? You have some data there, just not much.

Line 209: what are "vacant vocal states"?

Line 277: I think you mean "Assuming a natural trend to develop high diversity songs either via imitation or improvisation"??

Figure 5: another very busy mega-figure. It's likely not all of this is necessary. 5e is perhaps difficult to interpret and not necessary. In a-d, why represent each pair with 10 non-independent points? What data were used for the analysis? Also the figure caption seems to run out of text or the text has been cut off at the end. Subfigure i is indicated as (i) whereas for others it is in bold. In the figure, letters h and i are lost anyway.

Figure 6: the caption seems to be truncated.

Discussion

Line 452: insert "in" after "abundant"; "pupils" should be "pupil's "

Line 473: insert "to" after "due"

Lines 459-460: You call this phenomenon "balanced imitation" but I don't think that you've presented any evidence that the new syllables included by tutors when imitating poor tutors are imitative. Sure, they're adding new syllables, but where from? Are they imitations or have they developed them de novo, perhaps from some innate template? If they are imitative, who have they imitated? Early in the MS you dismiss that they're imitating neighbouring tutors although I would find that the most parsimonious explanation. Indeed, dismissing this probably requires more evidence than presented in the MS at this point. Anyway my point is if it really is "balanced imitation" they have to be imitating someone, but who? If not, then it's a poor term and maybe should be "balanced vocal production" or "balanced song production"?

Line 503: insert "Australian" before "zebra" at end of sentence.

Para 496 – 503: This is very speculative. On the one hand you argue that less variability in songs is due to weaker balanced imitation, but it could also be due to stronger imitation as you point out earlier in the MS.

Lines 519-521: On splitting being rare: as mentioned earlier, this is likely an artefact of your methods. If they split a syllable, you would not have matched it and would not record that whereas when they combine full syllables, they are matched to existing syllables and detected. This needs revision.

Methods

Lines 564-567: More details required. Measurement of similarity in acoustics is complex and can be done many ways. Just saying it was done using a program with “standard settings” does not provide nearly enough information to have any idea how this was done. What is the underlying method used? What are the relevant settings?

Line 570: Link to Sound Analysis Pro doesn't work. In fact I can't find any working webpage for this software. There is therefore no information about feature measurement in the 10ms windows.

Line 572: 50dB relative to what? dB are not absolute units and have little meaning without a reference value.

Line 574: Pitch vs frequency. See notes above about this. If you used frequency then say so; if you logged it first then say this and use pitch.

Line 575: There's no heat map in Fig 1b. Assuming you mean the heatmaps in Fig 3, you can't get FM from a single 1 or even 10ms slice – you'd need a greater duration than that. This needs clarification and revision.

Lines 579-582: I was confused earlier why you used entropy here. Why not use simple proportion of sounds in each vocal state? Isn't that the relevant thing in your analysis? Entropy seems to take this a step further without adding anything. (See note above about being close to maximum entropy and this being a circular argument.)

Balanced imitation MS rebuttal

REVIEWER COMMENTS

We note that since we added a new Figure 1, all of the reviewers citations to figures in the paper are move up by 1.

Reviewer #1:

In this study, the authors track the cultural transmission of songs within a large multi-generational colony of zebra finches, and identify a previously unknown factor that shapes vocal culture. Zebra finches learn to sing as juveniles by imitating the songs of their (genetic or foster) fathers. It has been long known that learning outcomes are very variable, and this variability has been accounted for by variability in learning capacity and by innate species-specific biases (namely, pupils “correct” abnormal tutors towards a more species-typical song). However, here the authors show that independently of these two factors, imitation variability can be explained by pupils’ tendency to adjust the diversity of their tutor’s song, such that common elements become less frequent, and rare elements become more abundant. In a similar manner to balancing selection in evolution which maintains genotypic variability in populations, the authors conclude that a frequency-dependent imitation bias in zebra finches maintains a “steady state” of relatively high song diversity through learning generations, and could explain the fact that zebra finch populations don’t converge to local dialects. These findings are very interesting, and are likely to open a fruitful new direction in cultural evolution studies.

The study is well conducted, and the analysis methods are adequate and innovative. In particular, the authors extract a relatively small number of distinct short-time-scale (10 ms) acoustic “building blocks” that characterize the songs repertoire of birds in their colony (and likely, of zebra finch songs in general), and use these to quantify the information entropy (diversity) of songs. This is important, since there is growing evidence that these shorter time scales might be more important to the birds than time scales that are more salient to the human ear, like syllables and motifs. In addition, the data of song samples in identified tutor-pupil families within a colony are made available in a public song library at the Linguistic Data Consortium catalog at UPenn, which is hopefully the first step in building a large cross-study library of song cultures.

We thank the reviewer for this positive summary of the paper. We kept these aspects of the revised paper.

All the above make the manuscript a strong contribution that would be of interest to the broad readership of Nature Communications. However, I have several comments regarding the need to supply information about the diversity of tutor songs included in the study, to discuss alternative interpretations of the results, and to clarify some of the reasoning and methodology, as detailed below:

R1.1 We are grateful for the many suggestions and ideas for improving our MS, all of which were carefully addressed in the revised MS. Our revision includes additional statistical analysis, improved figures and better documentation of methods, and alternative interpretations as elaborated below.

General comments:

1. It would be good if the authors could provide some information about the overall diversity of tutor songs within their sample. Since at least some tutors are ex pupils of other tutors, it would be good to rule out the possibility that low-diversity or high-diversity tutors share some other common features besides acoustic diversity that may influence imitation.

R1.2 To address this concern, we tested if abundances of specific vocal states are similar across low-diversity or high-diversity tutor songs, which could potentially influence Imitation. We pooled together songs from tutors that had the lowest diversity (bottom quartile) and calculated the diversity of their “pooled song”. We found that the diversity increased from a mean of 2.99 bits to 3.17 bits in the pooled song, which is still low compared to the mean diversity in the top quartile (mean = 3.16 bits, pooled = 3.27 bits). The distribution of abundances of pooled vocal states explain this difference:

Figure R1.2: Vocal state abundances of low diversity tutor songs pooled together (lowest 10%) vs. pooled high diversity tutor songs (top 10%).

As opposed to the nearly flat distribution of vocal state abundances in the high diversity songs, low diversity songs tend to have higher proportion of states 9 & 10, which correspond to high pitch sounds. This is interesting because in this respect, the low diversity songs are structurally similar to isolate songs, which are often of higher pitch.

Is there a similarity in song features of tutors that were not well imitated (and tutors who were) aside from the diversity?

R1.3 Yes, see R1.2 above. However high pitch syllables, which are more common in the low diversity songs are often imitated well.

2. The authors present the balancing selection employed by pupils with regard to their tutors' song as an adaptation, that "sustains cross-generational homeostasis that prevents the collapse of vocal culture into either complete uniformity or chaos." However, 1. It is not clear at this point whether this imitation phenotype is indeed adaptive, and the authors should consider alternative possibilities, such as that it is a by-product or side-effect of the birds' perceptual biases.

R1.4 We agree that we have not causally demonstrated this hypothesis. This was a proposed hypothesis to explain the results. We also agree with the reviewer that, whether adaptive or not, the mechanism is likely to involve perceptual biases. More importantly, even if balanced imitation is not adaptive in domesticated zebra finches, it could be a pre-adaptation in other species and perhaps in other forms of cultures. As suggested by the reviewer, we proposed alternative hypothesis, in the last paragraph of the discussion where we discuss adaption:

"Previous studies^{25,37} suggest that high song diversity in a colony of zebra finches could be adaptive. In the zebra finch female, dopamine response to songs is tuned to the song of her mate³⁷. To the extent that balanced imitation can also sustain the acoustic diversity of songs within a colony, it might also enable the females to respond selectively to their mates. Balanced imitation is also of interest in a broader context of vocal and non-vocal cultures in humans. In general, cultures may vary in their stability and in their richness (polymorphism), and balanced imitation could potentially explain how different morphs of cultures come about. At the population level, balanced imitation can be thought of as an example of a balancing (negative frequency-dependent) selection of morphs, which can promote polymorphism by preventing the extinction of rare morphs. At the individual level, it can be thought of as a mechanism that promote diversity in the skills that are acquired. It would be particularly interesting to test how imitation biases might interact with the structure (topology) of communication networks, in determining how cultural behavior spreads and filtered over space and time³⁸. Finally, other possible mechanisms could potentially explain balanced imitation including perceptual biases³⁹ and habituation⁴⁰."

2. The adaptive value the authors attribute to balanced imitation is not clearly explained: why is cross-generational homeostasis important? What is the problem with a vocal culture "collapsing into chaos", and what is meant by "chaos"?

R1.5 see R1.4 above. We consider cross-generational homeostasis of vocal cultures important because anything that is allowed to diversify unchecked, could eventually limit the ability of species recognition and communication. By "chaos" we mean that vocal

cultures can be unstable and semi-random. The consequences could include preventing the accumulation of knowledge. We now provide a more detailed explanation.

3. It seems that there is something special about zebra finches compared to other species that have song dialects – how do the authors account for it given their results? Namely, if balanced imitation is what prevents zebra finch vocal culture collapsing into total uniformity, how could species that have song dialects avoid this fate?

R1.6 This is an important question that our revised discussion now addresses. We suggest that the mechanism we observe here could explain what we see in other species. In species with weaker balanced imitation, we are likely to see more convergence, and perhaps in some parameter space this would generate dialects. Even then, the balancing effect is likely to act against total uniformity.

3. Overall, the writing is clear and succinct, but in some parts I struggled to follow the authors' reasoning, and some of the method descriptions were too succinct (at least for me) and need to be more comprehensively explained. These are detailed in the specific comments below.

Specific comments:

Some parts of the abstract could be made clearer, as follows:

Abstract line 18-19: even in cases of accurate imitation, pupils often recombined imitated syllables to form new units, and, therefore, distributions of syllable types in pupils' songs were not well correlated with their tutors'; then why are these considered cases of accurate imitation? It's not clear how this explains good song imitation in some families and bad imitation in others.

R1.7 By accurate imitation we mean that all the vocal sound in tutor's song were copied. The word accurate is misleading because we mean accurate at the phonology level. We changed the wording to:

“even in cases of nearly complete acoustic structure imitation, pupils often recombined copied syllables to form new units, and, therefore, distributions of syllable types in pupils' songs were not correlated with their tutors”

Abstract: lines 20-23: ... rare vocalizations in tutors' songs became more abundant in their pupils' songs, and vice versa. Consequently, cultural transmission of tutor songs that were high in acoustic diversity were stronger than those that were low in diversity. This second sentence seems to contradict the “vice versa” in the previous sentence – if pupils always balanced out tutor's diversity, why was cultural transmission stronger in higher diversity tutors?

R1.8 What we meant to say is that low diversity leads to biased imitation and improvisations, which resulted in weaker cultural transmission. We revised the sentence to:

“Consequently, the imitation of tutor songs that were low in acoustic diversity were biased, with more improvisations, and therefore cultural transmission was weaker compared to tutored songs that were high in diversity.”

Last sentence of abstract: “... that prevents the collapse of vocal culture into either complete uniformity or chaos.” Very dramatic, but vague. Sounds like something terrible is going to happen if there is no balanced imitation, but what exactly? (see general comment #2#)

R1.9 We would like to retain this statement, which we now elaborate upon in the revised discussion (see last paragraphs in the revised discussion).

Introduction:

Page 2 lines 13: at this point it is not clear what exactly is meant by acoustic diversity: is it the variability across tutor song renditions, or the range of acoustic states within a single song, or something else? This makes the remainder of the paragraph a bit cryptic. This of course becomes clearer after on, but since acoustic diversity is such a central measure in the study, it could help if the authors provide a brief explanation when the term is first mentioned.

R1.10 In the revised MS, we have now provided a brief explanation on first the mention of acoustic diversity: “We tested how a rich (polymorphic) repertoire of song syllables is sustained during cultural transmission²⁵ in zebra finch. We estimate song complexity at a high level of song “repertoire”, based on the diversity of vocalization “types” produced. We quantified song polymorphism using measures of acoustic diversity, for studying the statistics of song imitation in a large colony.”

Results:

Lines 26-27 (page 2): how did the authors determine there was no imitation across families? This is not explained in the methods

R1.11 Our initial analyses were anecdotal of case-by-case examples we examined. We now performed a quantitative analysis, and tested if pupil's who imitated their tutor's song poorly were influenced by the song of other tutors in their room, whom they could hear:

“We tested if pupil's who imitated their tutor's song poorly were influenced by the song of other tutors in their room, whom they could hear. We did not record the entire acoustic environments that existed in each room in each developmental period. However, we found 22 clutches with pupils who imitated poorly (lowest quartile), where we have recordings from at least one neighboring tutor that raised chicks in the same room at the same time. In cases of more than a single neighbor tutor, we selected the tutor that was imitated most accurately by his sons. We then measured the similarity between the song

of pupils who imitated their tutor poorly, and the song of that ‘popular’ neighbor tutor. We performed this analysis in four different colony rooms. We compared these similarities to shuffled data: pairing each pupil with a non-neighbor tutor from a different room. We found that the mean similarity within a room was not higher than that across rooms (within a room: $38.7 \pm 2.6\%$; across rooms: $40.1 \pm 3.6\%$). Although we cannot exclude the possibility that these pupils might have copied song elements from tutors that we did not record, results are consistent with earlier observations, suggesting that zebra finches mainly imitate tutors that live with them and, in those conditions, ignore songs from birds they cannot interact with.”

Suppl. Fig 1: what is the y axis?

Add label ‘counts’

Fig 1a: does every “pupil” ellipse represent a single bird or a single clutch of a given tutor?

R1.12 Each ellipse (e.g. node) represents a single pupil. We now added symbol labels to indicate the clutches, and mentioned that each node is one individual animal. See revised figure 1a.

Page 3, lines 3-6: “In both kinds of cases, the birds still produced their syllables in the typical repeated song motif of 2-6 syllable types (Fig. 1b,c). This suggested to us that pupils might more accurately imitate tutor songs that are rich in acoustic structure (i.e., acoustically diverse), while improvising upon impoverished tutor songs.” I don’t understand the transition between the first and second sentence – how does the fact that pupils sang repetitive motifs suggest that they imitate tutors with rich songs more accurately?

R1.13 The paragraph was not well constructed. Changed to:

“For example, tutor Aq12 had a very simple song with one syllable type and all of his pupils imitated the syllable poorly. Some pupils introduced apparently novel syllable types in developing their own songs (Fig. 2b). In contrast, tutor DG1 had a more complex song, with five syllable types, and all of his pupils imitated the syllables and the sequence much more accurately, with little to no introduction of novel syllables (Fig. 2c). In both cases, pupils still produced their syllables in the typical zebra finch repeated song motifs of 2-6 syllable types per motif (Fig. 2b,c). This suggested to us that pupils might more accurately imitate tutor songs that are rich in acoustic structure (i.e., acoustically diverse), while improvising upon impoverished tutor songs.” To test our impoverished tutor song hypothesis quantitatively, we calculated the acoustic diversity of songs at the level of syllable types, which we call syllable type diversity. We compared the syllable type diversity of tutors songs to those of their pupils.”

Page 3, lines 6-7: “To test this hypothesis quantitatively, we compared the diversity of song syllable types in tutors and their pupils.” It would help if the authors specify here the predictions of their hypothesis regarding the correlation between tutor and pupil diversity.

R1.14 We now add the hypothesis and predictions based on the initial findings in the sonograms. *“This suggested to us that pupils might more accurately imitate tutor songs that are rich in acoustic structure (i.e., acoustically diverse), while improvising upon impoverished tutor songs. If the hypothesis were true, we would expect to find that as tutor’ song diversity decreases, then pupil’s imitation similarity should also decrease.”*

Later in the text, the authors express surprise at not finding a positive correlation between tutor and pupil syllable diversity, but why would they expect to find a positive correlation (see point on that below)? It rather seems that their hypothesis suggests there should be an interaction between diversity and imitation accuracy.

R1.15 The reviewer is correct. We have modified the text, as answered in a related comment above, and now test for correlations between tutor song diversity and pupil imitation error (as above).

Page 3, line 7 – up until now, the authors use the terms simple, complex rich in acoustic structure, and diversity without explaining what they mean.

R1.16 By simple and complex song acoustic structure, we are referring to the level of diversity of syllable types based on acoustic features (FM, pitch, duration) of the syllables in the birds song motif. We quantify that diversity using entropy ($-\sum p_i (\log_2(p_i))$). We have now explained this in more detail in the revised paper. Overall: we used the term *acoustic diversity* as a generic term (before specifying it technically). Then, we use two different estimates of acoustic diversity: one at the syllable level, which we call ‘*syllable type diversity*’ and then at the vocal states level, which we call ‘*song diversity*’. We revised the first statement to:

“... we calculated the acoustic diversity of songs at the level of syllable types, which we call syllable type diversity. We compared the syllable type diversity of tutors songs to those of their pupils.”

Then:

“Similar to syllable and transition diversity, for each bird’s song, we calculated acoustic diversity over the 10 vocal states using Shannon information entropy, $-\sum p_i (\log_2(p_i))$, but here p_i is the proportion of sounds within each vocal state i . Hereafter we will refer to this measure as ‘*song diversity*’.”

These terms are now used consistently throughout the revised MS.

Page 4, lines 6-7: 1. how was syllable relative frequency (abundance) calculated? Was is the proportion of performing a certain syllable type out of all the syllables in a single song bout? Song motif? Entire song sample of a bird?

R1.17 The syllable relative frequencies were calculated by performing cluster analysis of the entire song sample obtained for each bird: for each bird, we sampled hundreds of song bouts containing at least 3,000 syllables in total, which we then automatically clustered. Our clustering method then identified clusters (syllable types) and automatically

counted how many syllables appeared in each cluster. The relative frequencies were calculated from these counts. We added a new Methods section (Analysis of syllable type diversity:) to properly document it.

2. How did the authors determine that two different birds share a syllable?

R1.18 We added a Methods section to explain this:

“Evaluation of syllable type recombination: Our measure of syllable type diversity is based on classifying syllable types within each bird, and it does not evaluate or compare syllable types across birds. For evaluating syllable type recombination, however, we need to determine which syllables were copied, either as a unit or in parts. Using similarity measurement with Sound Analysis Pro, for each syllable in tutor song we automatically detected sections of similarity in pupil’s song. Evaluating these sections allowed us to determine if the boundaries are consistent across tutor’s and pupil’s songs. In particular we could detect splitting (a similarity section with one interval in tutor song and two disjoint intervals in pupil’s song) or merging (a similarity section with one interval in pupil song and two disjoint intervals in tutor’s song). Note, however, that we only included cases where imitation was subjectively apparent, which is easier to determine in cases of merging compared to splitting. Therefore, our lower bound estimates could be biased toward the detection of merging.”

Page 4, lines 8-10. The study’s findings center on measuring song diversity using Shannon’s information entropy. Since this measure was not used before to quantify diversity of vocalizations, or motor sequences in general (if it was, the authors don’t cite any), it would be good to provide more detailed information for the reader, and preferably also some means to intuitively assess what is a high entropy or low entropy song.

R1.19 We added an explanation: “We then calculated the relative frequency (abundance) of each syllable type and used Shannon information entropy²⁷ to estimate the diversity of song syllable types produced by each bird. Specifically, for each bird’s song, we calculated the proportion p_i of syllables produced for each syllable type i , and computed $-\sum p_i (\log_2(p_i))$, as an entropy estimate of syllable type diversity. The Shannon information entropy is a better estimate of diversity compared to just counting syllable types because it considers the frequencies (abundances) of each type. We then tested if the syllable type diversity was correlated in tutors and their pupils.”

Supp fig 2: both axes labels are missing.

Fixed

Page 4, line 11: why is it surprising that there is lack of correlation between tutor and pupil diversity? Not all pupils copied the songs accurately, so why are those expected to be as diverse as their tutors? According to the hypothesis presented by the examples in Fig 1b-c, the authors would expect that only the diversities of pupils with high-diversity tutors would be correlated with their tutors’ diversity, correct?

See R1.14, and R1.15

Page 4, line 23: rearrangement of tutor syllables should not affect the syllable diversity scores and relative frequencies of syllables, only merging of tutor syllables would.

R1.20 This is correct. We now only use the term 'recombine', referring to the actual units such as merging, and splitting.

Page 4, lines 23-25: "We assessed a lower bound estimate of similarity in the syllable boundaries of tutor and pupil songs ". This sentence is cryptic to me - what is meant by lower bound estimate? Did the authors specifically assess syllables that were merged? Do they mean that they restricted their analysis to syllables whose boundaries did not match those of the tutor but that were clearly imitated? The selection criteria for syllable assessment should be more clearly explained.

R1.21 By lower bound estimate of similarity, we mean cases where syllable imitation could be detected by Sound Analysis Pro, with similarity section(s) that includes the entire tutor's syllable. We also required that a visual observation confirm that imitation is apparent. This method can detect both syllable merging and splitting. However, merging may be easier to confirm, which might have imposed a bias. We call it a lower bound estimate because the true number of deviations is likely to be much higher. This is now all described in the Methods, see "Evaluation of syllable type recombination"

Fig 3b: 3D plot looks 2D. I don't see the data on the entropy axis.

R1.22 We removed the 3D panel and added axes.

Page 6, lines 18-21, Fig 3c legend and methods (page 17, lines 34-36): how were the contours of the pitch distribution, and the cluster boundaries in each pitch slice identified? This is not clearly explained in the methods. Also, the methods refer to Supp Fig 1, but does not show pitch categories.

R1.23 This is now shown in the revised figure: The peaks in the pitch histograms were used for setting the specific categories of low to high pitch, and the minima points on each side of the peaks set as the boundaries. This is now explained in the main text.

Fig 3 f legend: all male birds recorded means pupils and tutors, or just tutors?
Fig 3g – what does median indicate, the median of what, diversity in all birds, tutors and pupils?

R1.24 All birds means both pupils and tutors, which we now specify in case other readers have the same question.

Page 6, lines 41-42: "However, at least, in the top two quartiles the coefficient of determination seems robust ($R^2 = 0.23$, Fig. 3e) with no apparent interaction with specific family branches." Isn't that trivial? If the imitation is good (songs are similar) then the diversity between tutor and pupil should also be similar. In general, I don't understand the

point that panel e is intended to make. It seems trivial that the correlation between tutor and pupil diversity in bad pupils will be less robust than in good pupils.

R1.25 Yes, this is expected. But we don't consider it trivial. It supports our hypothesis about a relationship between tutor song diversity and imitation accuracy. We have now revised the text to indicate this. The correlations in bad pupils is a bit lower but comparable to that of good pupil. The only reason we show them separately is because that correlation might be driven by one family branch of bad tutors (with song diversity at about 2.6). So we just wanted to show that there is correlation in both good and bad imitation, and also when pooling all birds together.

Is the point of the panel just to show the advantage of looking at diversity across vocal states instead of syllables?

R1.26. Yes. The lack of correlation at the syllable level means that it does not capture cultural transmission at that level. The success at the vocal state level makes it possible to proceed.

Page 7, lines 9-10: "...such that a large proportion of pupils with low song diversity had tutors with high song diversity, and vice versa." What is meant by a "large" proportion?

R1.27. 42% (sentence revised accordingly).

Fig 2g: not stated whether the r value is significant.

R1.28 It is significant, but we presented it as part of our fixed effect model, so it would have been redundant to show it as a separate test: "Results confirmed that both factors contribute about equally to pupil song diversity (imitation similarity: $t=5.0$, $p=1.4e-6$; tutor song diversity: $t=4.6$, $p=7.9e-6$)."

Page 8, lines 27-28: I don't follow the reasoning: how do the authors know that low similarity was a result of poor imitation in this case, but a result of balancing selection in Fig 4 b and c?

R1.29 This is an important point that we did not emphasized enough: Balanced imitation is driven by low diversity in tutor song – not in pupil's song. In contrast, poor imitation is correlated with low diversity in pupil's song and not in tutor's song. We clarified this point in the revised MS.

Page 8, lines 41-42: "Assuming a natural trend to develop low diversity songs either via imitation or improvisation,...". "low" = "high"?

Fixed typo, should be high.

Page 8, lines 43-44: "We found that pupils that imitated poorly, regardless of tutor song diversity, tended to have low diversity songs ($R^2=0.20$)."

What correlation does the R^2 value refer to?

This value refers to the correlation in Fig 4g, which we now cite.

Page 9, lines 6-7: “Pupils of tutors with high song diversity who imitate well, produce songs of comparably high diversity.” Tautological sentence: how can the pupils not produce high diversity songs if they imitate high diversity tutors well?

True. Changed to “As expected...”

Page 10, lines 13-14: “to test our hypothesis farther...’ which hypothesis is referred to?

R1.30 Statement removed, hypothesis is not necessary here.

Page 10, lines 14-15: why are abundances called raw frequencies? Is this the same abundance calculation that the diversity measure is based on?

R1.31 Abundances = frequencies, we removed ‘raw’.

Fig 5a: what does raw distribution mean?

R1.32 Replaced by ‘scatter plots’

Page 12, line 8: Blue lines, do you mean colored lines?

R1.33 Yes. We left this description as is, since it is common to mention the color of a object without adding the word “color” to every color description.

Page 10, lines 34: I don’t understand how fig 5e shows regression towards the mean. Would help if the authors could elaborate. Fig 5e, what does the dashed line indicate?

R1.34 We removed this panel and the statements about regression toward the mean, which are confusing and not necessary.

Page 10, lines 36-37: I don’t understand the gain calculation. What exactly is the median calculated over?

R1.36 We clarified that: “To evaluate if this was the case, we calculated the median abundance of pupil vocal states within 0.1 abundance bins in tutor’s song. We then calculated the abundance ratio for that median. For example, if at the window centered at 0.1 tutor abundance, and if the median pupil vocal state abundance was 0.2, then the gain ratio would be 2. A gain value of 1 (y-axis in Fig. 5f) represents identical abundance of all 10 vocal states in pupil and tutor. A gain value of 2 indicates a doubling of abundances in the pupil (amplification), and a value of 0.5 halving (attenuation).”

Page 13, lines 8-9: “Can we predict imitation outcomes based on the mean features of a tutor song?” I couldn’t follow the rationale for asking this question. What hypothesis is tested here?

R1.37 Changed to: “We asked if we can we predict imitation outcomes based on the mean features of a tutor song? If songs of low diversity were culturally transmitted less than

high diversity songs, than songs with extreme mean features – which are typically of low diversity – should be transmitted less. To evaluate this...”

Page 13, lines 22-25: “Across the three other colonies, the distribution of mean song features was to a large extent confined within the range of good imitations in our colony. Therefore, the range of mean feature values of best imitations, but not of worst imitations, seems consistent across zebra finch colonies.” This is unclear to me. How can we distinguish in Fig 6d-f between good and bad imitators in the other 3 colonies? The plots show only the mean feature values for each bird. How can the authors deduce that the birds in the 3 additional colonies that were within the range of good imitation in the first colony were also good imitators?

R1.38 We are not claiming whether birds in the other colonies were good imitators. All we want to test is if the range of song features that we see in other colonies is similar to that of good imitation in our colony. The only argument is that, over generations, songs of high feature diversity are more influential, and therefore shape the overall distribution of mean song features. We revised the statement to “the range of mean feature values of best imitations in our colony...”

Last paragraph of the results: I am not absolutely clear about the significance of the finding that balanced imitation reduces the frequency of songs with extreme mean feature values. It is not mentioned in the discussion. Is the point of this analysis to demonstrate balanced imitation across colonies? If that’s the case, it might help to change the paragraph’s title accordingly.

R1.39 We cannot make this claim. All we can say is that this outcome is consistent with the notion that over generations, songs of high feature diversity are more influential, and therefore shape the overall distribution of mean song features in a similar manner across colonies.

Discussion: Page 15, lines 16-17: which of the findings suggest that the imitation bias is very sensitive to fluctuations in abundance?

R1.40 It is the gain plots in Fig. 5f. We can see that even small deviations from the even ratio has an effect. A reference to the figure has been added to the discussion.

Page 15, line 20: “...or diverging too much into chaos.” See general comment #2 – the adaptive value the authors impute to balanced imitation needs to be clarified.

R1.41 we revised to clarify: “We suggest that balanced imitation prevents vocal cultural learning from converging too much into complete uniformity or diverging too much into chaos. In either occurs, the communication system might become deficient: high song uniformity could reduce individual identity, whereas a chaotic song culture might reduce group identity.”

Page 15, lines 31-33: “The current study confirms that a proportion of pupils of high diversity tutors acquired very low diversity songs, perhaps due social inhibition of song imitation.” But it could also be due to lower capacity for imitation.

R1.42 Agreed. We changed to: “The current study confirms that a proportion of pupils of high diversity tutors acquired very low diversity songs, perhaps due to social inhibition of song imitation, or due to lower genetic capacity for imitation”

Page 15, lines 9: Could the authors speculate on which ecological conditions would favor more or less inter-individual variability (see again general comment #2)?

R1.43 We added to this paragraph more speculations about ecological conditions. We suggest that more inter-individual variability would be favored by higher density populations within species, and less variability would be favored by low density within species or high density between species. The population density would presumably influence the abundance of vocal states heard, and thus influence what is learned.

Minor corrections:

Refs 15 and 19 are duplicated.

Fixed

Page 2 line 31: ref 27 should be ref 28? Maybe I’m wrong. Does Shannon talk about similarity or just entropy? Same for Page 4 line 5, and other parts of the text.

Fixed

Reviewer #2:

Tchernichovski et al. ask how vocal diversity is maintained in a colony of zebra finches. Based on previous work it seems that a colony would converge onto a narrow range of syllables and song complexity. However, this is not the case. The authors analyzed song in a colony to determine how diversity is maintained. They develop methods to analyze ‘acoustic diversity’ in the colony. They find diversity within a colony is maintained through ‘balanced imitation’ in which the song of some tutors is well-copied but the acoustic diversity decreases in the pupil, and the tutor songs that are poorly copied increase acoustic diversity.

This is a well-written paper in which the authors take us on a scientific journey of discovery. The ideas are interesting, novel, and noteworthy. The data analysis will be an important contribution to the field and has the potential to be used for by many labs (as other analysis from the OT lab).

We thank the reviewer for these positive and kind comments. We address all issues as shown below.

I have a few comments for consideration:

1. I was a little confused how the authors define 'diversity'. They discuss acoustic diversity, song diversity, syllable diversity, and vocal state diversity. Are these all the same? In the methods there is a definition of diversity, but it is more statistical, could the authors provide a more intuitive description of 'diversity'?

R2.1 We addressed this issue: We now use the term *acoustic diversity* as a generic term (before specifying it technically). Then, we use two different estimates of acoustic diversity: one at the syllable level (that failed), which we call '*syllable type diversity*' and then at the vocal states level (which succeeded), which we call '*song diversity*'. To clarify this, we revised the first statement to:

"To test this hypothesis quantitatively, we calculated the acoustic diversity of songs at the level of syllable types, which we call syllable type diversity. We compared the syllable type diversity of tutors songs to those of their pupils."

Then:

"Similar to syllable and transition diversity, for each bird's song, we calculated acoustic diversity over the 10 vocal states using Shannon information entropy, $-\sum p_i (\log_2(p_i))$, but here p_i is the proportion of sounds within each vocal state i . Hereafter we will refer to this measure as 'song diversity'."

These terms are now used consistently throughout the revised MS.

2. Figure 4. The authors state that there are correlations in Fig. 4e and g. However, it looks like much of the correlation is driven by one or two tutors. That is, if data points less than 2.8 song diversity are removed, it looks like the correlation would be lost. Is that the case? If so, how can those results be interpreted? (Is this a concern?)

R2.2 These tutors explain about half of the correlation. Removing these data decreases correlation (R) from 0.5 to 0.24, but did not eliminate the statistical significance of the correlation ($p = 0.003$). We have now mentioned this results in the revised paper (although did not include the graphs as shown below).

With the two tutors:

Without the two tutors:

3. It would be interesting to have the authors suggest how these ideas could impact other research, outside of songbirds. It seems these ideas are important for other areas of research and it would be fun to propose some additional impacts of this work.

R2.3 We thank the reviewer for making this suggestion. We added this paragraph to the discussion:

“Previous studies^{25,37} suggest that high song diversity in a colony of zebra finches could be adaptive. In the zebra finch female, dopamine response to songs is tuned to the song of her mate³⁷. To the extent that balanced imitation can also sustain the acoustic diversity of songs within a colony, it might also enable the females to respond selectively to their mates. Balanced imitation is also of interest in a broader context of vocal and non-vocal cultures in humans. In general, cultures may vary in their stability and in their richness (polymorphism), and balanced imitation could potentially explain how different morphs of cultures come about. At the population level, balanced imitation can be thought of as an example of a balancing (negative frequency-dependent) selection of morphs, which can promote polymorphism by preventing the extinction of rare morphs. At the individual level, it can be thought of as a mechanism that promote diversity in the skills that are acquired. It would be particularly interesting to test how imitation biases might interact with the structure (topology) of communication networks, in determining how cultural behavior spreads and filtered over space and time³⁸. Finally, other possible mechanisms could potentially explain balanced imitation including perceptual biases³⁹ and habituation⁴⁰.”

4. One of the findings that is particularly interesting and powerful is the comparison to other colonies. This is part of the final paragraph in the results, but perhaps should be highlighted more.

R2.4 We agree with the reviewer that analysis of song culture across colonies is potentially powerful and promising approach. We expanded discussion of these results, to more clearly state what we can conclude and are not able to. We thank the reviewer for the suggestion.

Very minor points:

5. Line 129. What is Shannon information? Could the authors provide a sentence or two what this analysis is, and why it is useful? I do not have a strong background in statistics, so it remained mysterious to me. Adding this information would be good for a more general reader.

R2.5 We used Shannon information as simple a diversity measure, in a manner that has little to do with Shannon's ideas about information uncertainty. The measure weighs each vocal element (syllable or vocal state) by its abundance, and presents the entropy (diversity) of the distribution in units of bits. Although other diversity measures would have worked, the advantage of Shannon information is that we could then use the exact same measure to look at acoustic (syllable or vocal state) and transition entropy (song syntax). We expanded the description in the main text:

"We then calculated the relative frequency (abundance) of each syllable type and used Shannon information entropy²⁷ to estimate the diversity of song syllable types produced by each bird. The measure weighs each vocal element (syllable or vocal state) by its abundance, and presents the entropy (diversity) of the distribution in units of bits. We then used the same Shannon information measure to evaluate syllable transition entropy (song syntax). Specifically, for each bird's song, we calculated the proportion p_i of syllables produced for each syllable type i , and computed $-\sum p_i (\log_2(p_i))$, as an entropy estimate of syllable type diversity. The Shannon information entropy is a better estimate of diversity compared to just counting syllable types because it considers the frequencies (abundances) of each type. The more syllable types produced, and the more even their abundances are, the higher the entropy. We then tested if the syllable type diversity was correlated in tutors and their pupils."

6. On line 83-84. What is the range of pupils per tutor? This can potentially weight the data and interpretation, as the authors mention later in the paper.

R2.6 We now include in Figure 1 information about clutch members. This way one can visualize how many of the pupils of each tutor were clutch mates.

7. Line 258. "all of the grand-pupils imitated more accurately." Than the father, or the grand-father? This is a little unclear as written.

R2.7: changed to "For example, in the two lineages (HP10 and DG4) with the greatest number of first-generation pupils that imitated poorly, all of the second generation (grand) pupils imitated the song of their father more accurately than the father imitated of the grandfather."

8. Line 259. "the grand-tutor songs were atypical." Atypical compared to what?

R2.8: These grand-tutors songs were unbalanced: one with high proportion of high pitch sounds and the other with high proportion of unmodulated harmonic stacks. We revised the sentence to indicate this.

9. Line 280. Add reference to Fig. 4g (missing in the text).

Fixed

10. Line 315. “To test our hypothesis further...” it was not clear what the hypothesis was at this point.

Removed

11. Last paragraph of results. It is worth reminding the reader that the data analyzed are from RU2019. It seems that RU2002 is from an earlier time in the same colony, which is interesting to see how the colony changed (or did not change) over time. The author state “Across the three other colonies,...” Should this be two other colonies, since RU2002 is technically the same colony? This was a little confusing.

R2.9: We consider the RU2019 and RU2002 colonies as fairly independent: In 2017 approximately 300 birds brought from Duke University were mixed with approximately 150 birds that were descendants of the RU2002 Millbrook colony. Recordings are also 15 years apart. This is now mentioned in the revised manuscript

12. Line 484. “Our statistical analysis suggest” – suggests

Fixed

13. Line 503. Add ‘Australian’ to zebra finches.

Fixed

Reviewer #3:

This MS presents a detailed analysis of similarities and differences between the songs of tutor and pupil zebra finches. The authors have used a large number of tutor-pupil pairs in the analysis, using multiple metrics and techniques to quantify and compare song learning outcomes. They show convincingly that pupils tend to normalize their songs, including new material when they have relatively poor tutors, here defined as tutors with a small repertoire compared with the norm. In terms of the amount of data, quantification and analysis, the MS is impressive. Its only real weakness in these aspects is a lack of detail on how sounds were measured and, in particular, compared quantitatively to derive measures of similarity that are central to the work. Techniques used are likely valid, but cannot be assessed with the methods presented.

R3.1 We thank the reviewer for the positive assessment and for the useful suggestions, which we address below. In particular, we performed several additional statistical

analyses, improved the figures and added more details about methods and similarity assessment as suggested.

The authors conclude that the pupils tend to 'balance' their songs and name this process "balanced imitation". While this is a valid theory, I have some comments/ concerns. One is that when pupils add new syllables to their songs, this term suggests that they imitate these syllables, but where from? The authors discount early on any learning from neighboring tutors. This assertion however is weak - the birds can hear each other and there is no evidence provided that this does not occur.

R3.2 Reviewer 1 had a similar concern, which we addressed as follows:

"We tested if pupil's who imitated their tutor's song poorly were influenced by the song of other tutors in their room, whom they could hear. We did not record the entire acoustic environments that existed in each room in each developmental period. However, we found 22 clutches with pupils who imitated poorly (lowest quartile), where we have recordings from at least one neighboring tutor that raised chicks in the same room at the same time. In cases of more than a single neighbor tutor, we selected the tutor that was imitated most accurately by his sons. We then measured the similarity between the song of pupils who imitated their tutor poorly, and the song of that 'popular' neighbor tutor. We performed this analysis in four different colony rooms. We compared these similarities to shuffled data: pairing each pupil with a non-neighbor tutor from a different room. We found that the mean similarity within a room was not higher than that across rooms (within a room: $38.7 \pm 2.6\%$; across rooms: $40.1 \pm 3.6\%$). Although we cannot exclude the possibility that these pupils might have copied song elements from tutors that we did not record, results are consistent with earlier observations, suggesting that zebra finches mainly imitate tutors that live with them and, in those conditions, ignore songs from birds they cannot interact with." See Suppl. 1 of the revised MS.

If the authors are correct though and there is no imitation from neighbors, then presumably the pupils are developing the syllables de novo with some innate syllable template. If this is the case then "balanced imitation" is a misnomer.

R3.3 We call "balanced imitation" the trend to develop high diversity songs either via imitation or improvisation. We think that we have evidence to both. Pupils of low diversity tutors appear to develop some of their syllables de novo, like isolate birds do. We do not think there is an exact innate template otherwise all such juveniles, or isolate birds would generate the same de novo syllables. Instead we consider this more like improvisation, where each bird can generate something different based on their experience or other factors, including genetic. We agree that developing de novo (innate/random) syllable types alone is not sufficient evidence to balanced imitation. We followed the reviewer suggestion and tested balanced imitation statistically (see R3.4 below). What we consider balanced imitation is the increase in song diversity, which we observe most strongly in cases of best imitations where tutored syllables were copied with little or no addition of new song elements. In these cases, balance was achieved exclusively via a biased (frequency dependent) copying of tutor's vocal sounds. We are not sure, however, if improvisation is also statistically biased and there is no easy way to test for this in our

data because improvisation is not as well defined as imitation is. We added this explanation to the main text.

My other concern is that the authors present "balanced imitation" as an active process. What is the evidence though this is not just a random process? That pupils listen to their tutors to some extent and just randomly add things if the tutor's song is not that interesting? If a species maintains a certain level of diversity in its songs (as demonstrated nicely by the authors), then it seems reasonable to assume that there is an underlying innate level of diversity that they use/learn/prefer. Perhaps this tendency to a certain diversity is not random and perhaps it is driven by a desire to have 'balance', but I do not think that the authors have demonstrated this convincingly. They must be able to refute the null (random) hypothesis.

R3.4 To address whether deviations in pupils' imitation are random, particular with low diversity tutors, we performed additional statistical analysis: our null hypothesis is that when the abundance of a vocal state in the tutor's song is high, his pupil is not more likely than chance to deviate from the model in a manner that produces a lower abundance of the corresponding vocal state in his song. In other words, if deviations (imitation 'errors') are random, then the likelihood of those errors to increase or decrease vocal diversity should be determined by the overall distribution of errors in our sample. In a previous study (Feher et al, 2009), some of us presented evidence that imitation from isolated tutors is biased: syllables with high abundance (20% or higher) in abnormal isolate tutor song were often less abundant in pupil's songs. Here we tested statistically if vocal states that are highly abundant in normal songs are imitated in a balanced manner, using the same 20% threshold:

As shown above, the distribution of tutor vs. pupil vocal state abundances is asymmetric: when tutor's vocal state abundance is above 20%, about 14% of corresponding pupil's states are above diagonal (hence decreasing song diversity). But looking in reverse: when pupil's vocal state is above 20%, a higher proportion of corresponding tutor's states (23%) are to the right of the diagonal. We found that this bias is statistically significant: To overcome dependencies between vocal states, we treated each tutor-pupil pair as a statistic. We randomly shuffled the direction tutor->pupil vs. pupil->tutor (without breaking the pairs) to obtain a random distribution of biases. We found that the observed bias to increase vocal diversity (namely in the direction that decrease abundance of vocal states that are already of high abundance) is higher than expected by chance (bootstrap direct p value = 0.032). Importantly, this bias is particularly strong in the top (most accurate) imitation quartile, where only one vocal state with tutor abundance > 20% is above the diagonal, as opposed to 11 cases below the diagonal (Fig. 6a). Therefore, evidence suggest that the bias we observed is associated with imitation, and is not entirely random, as we also observed in several specific cases (Fig. 6h-i).

The MS itself is very well written with excellent use of language. It is clear (if, at times, complex) and highly readable. The figures are excellent and provide a wealth of information, but at times become too small to read, and it is likely that parts can be removed. They are very good and thorough, but perhaps overly-thorough.

R3.5 We revised the figures, but we would like to keep the raw data panels in place for clarity of presentation.

Abstract

Excellent, v clear and easily understood.

Introduction

Line 53: "more variable across individuals"; Slightly confusing. Each individual has a high variability, or the individuals are different to each other? In other words, does this mean variability within or among individuals?

R3.6 changed to "much more variable among individuals".

Results

Line 75: Visually but not acoustically isolated from other families, so song learning across families cannot be ruled out. Found "no evidence of song imitation across families"; But would this not be difficult to detect due to variability?

See response R3.2 above.

Line 76: So were the birds removed and placed in isolation for a week or so for recording?

R3.7 Yes, this were alone for the week during recordings. We clarified this further in "Audio recordings" Methods.

Figure 1a and text lines 84-86: So you used mean clutch similarity rather than individual sibling similarity within the clutch which seems reasonable. But then Fig 1a is a little misleading as it only indicated the individual offspring with no indication of which were from the same clutch. While it's a nice figure, it's displaying information that is perhaps not relevant or, at worst, confusing as the individual was not necessarily the basis for analysis. Additionally there is no way of knowing which of the pupils were clutch mates and so no way of visually assessing how this might affect individual sibling learning.

R3.8 We added shapes to the figure to indicate pupils that belong to the same clutch:

See Figure 1 in the revised MS. We also think it is valuable to show both the individual data and the average data, and some journals now require.

Line 132-134: Is the Shannon entropy measure used as the measure of syllable type diversity?

R3.9 Yes. We revised the sentence to make this clearer.

So in fig 2c, are axes effectively the result of the entropy equation given in line 132? It's not clear. A simple measure of diversity is just mean number of syllable types produced per song which might also make sense as the values of the axes of fig 2c. Please clarify in text and figure.

R3.10 Yes, the axis in Fig. 2c the values determined from the entropy equation. Information entropy is a better estimate of diversity compared to just the mean number of syllable types because it considers the frequencies of each type. This way, rare syllable types do not bias the estimate. Another advantage is that the exact same entropy equation can be used to measure vocal state diversity and syllable syntax diversity (which other studies used as well). We now give this additional clarification in the paper, including why it is better than using the mean.

Line 150-151: You say that you restricted this analysis to where there was full imitation but the boundaries had changed. You make the point that all cases involved merging of syllables rather than splitting. But is this not inevitable if you restricted this to cases where the syllables could be identified? If the syllables were split, they wouldn't be recognizable as the same syllable any more. If this is the case, then the last sentence of this paragraph is a bit misleading as it suggests that splitting could have been seen but was not.

R3.11 Theoretically, since the manner in which we calculate similarity is continuous, and insensitive to syllable boundaries, if a copy of tutor's syllable is split or merged in pupil's

song, we should still be able to identify the two parts (or the merge) as copies. But we agree that splitting might be more difficult to detect. This is particularly true if the two parts are merged with other syllable types. This could possibly impose bias on our estimate, although we doubt if this can fully explain our observation. We added this caveat to the revised MS, in the Methods section "Evaluation of syllable type recombination".

Figure 2: Nice figures, but you don't indicate anywhere up to this point what "FM" and "Weiner entropy" are. You don't mention them in the caption and then, in the text, when you mention them you flick straight on to Fig 3. In addition, it's annoying having the y-axis labels for those two data sets in 2b hanging over the data from the previous data set. These graphs should be better separated (or deleted as you're really talking about them in Fig 3).

R3.12 We now define FM, Weiner entropy, and pitch on first use in the manuscript. We fixed the Y axis labels in Fig. 2 as suggested. Y axes are now properly separated. We also used a different example of tutor and pupil that show the merging effect in the context of the entire song motif.

Lines 173-175: A few things here need changing or explanation. Single FFT windows of only 10ms won't give information on frequency modulation; you need some longer time span to do this, so this doesn't make sense.

R3.13 This is done using multitaper spectral analysis. It is a technique that David Thompson developed at Bell Labs several years ago. In multi-taper spectral analysis, each FFT window (in this case 10ms) is calculated more than once, with a series of windowing functions. This allowed Thompson to compute frequency derivatives and time derivatives within each FFT window. Frequency modulation is the angle of this vector. The advantage here is that frequency modulation becomes a first order statistic of the spectral derivatives (so there is no need to compute the pitch contour first). see implementations in:

O Tchernichovski, F Nottebohm, CE Ho, B Pesaran, PP Mitra (2001) A procedure for an automated measurement of song similarity. *Animal behaviour* 59 (6), 1167-1176

Thomson, David J., and W. Fitzgerald. "Multitaper analysis of nonstationary and nonlinear time series data." *Nonlinear and nonstationary signal processing* (2000): 317-394.

Also, I hate to be a pedant, but you might be measuring frequency, not pitch? Pitch is the perception of frequency and is logarithmic. Looking at fig 3 the frequency scales on 3c are linear but for 3b are log (although I can barely read it).

R3.14 We are aware of this issue but there is no easy solution: zebra finch song is of broad power spectrum. So, there are many frequencies in each window. What we try to estimate is the periodicity. For a physicist, pitch is a scalar estimate of periodicity, whereas for a psychologist it is a percept. The YIN algorithm we use can be called either pitch or fundamental frequency estimate. At the time when we developed our sound analysis methods, our Bell Lab physicist collaborator strongly objected to using the term

'fundamental frequency' for the cepstral pitch estimate we used at the time. Fundamental frequency is well defined in physics, but is unlikely to be a valid term here given what we know about the mechanism of song production, which include two sources. So, we ended up using the term pitch, which is more loose. Both terms are used interchangeably in the literature for calculated acoustic features in different communities.

Line 176: "all birdsongs"? I suspect you mean all the songs of the birds in your study population. I initially thought you meant all species of song birds.

R3.15 Corrected to "all the songs in our sample"

Lines 183-184: inconsistent use of "&" and "and" between numbers.

Fixed

Line 198: remove comma after "least";

Fixed

Figure 3: although it's packed with useful information, the sub-figures are too small to be able to see details let alone read axes. X-axis of 3d in particular needs something. Suspect the dotted line is 0 with negative to the left and positive to the right?? 3b looks potentially interesting but can't see, on the face of it, clear reasons for the partitioning used. Might be better just to keep 3c and d? Also I'm not sure that f adds much as it's essentially the same data as in e and g - the long left tail.

R3.16 We followed reviewer suggestion and removed panels a and b on vocal state syllable clustering. We added units to the other panels. See revised Figure 3 in the MS.

Lines 204-207: Is this becoming circular? You used the existing songs to define the 10 vocal states, and then are pleasantly surprised that the median entropy is close to the maximum, i.e. surprised that there are lots of sounds in each of the vocal states that you used the sounds themselves to define. Why don't you have 12 states with 0 FM states between 1 and 2 and between 9 and 10? You have some data there, just not much.

R3.17 We thank the reviewer for making this point: We now clarify in the revised MS that this is not trivial:

"The distribution of song diversity remained stable over the lifespan of our colony (**Suppl. Fig. 2**). The highest theoretically possible diversity, with a uniform distribution of the 10 vocal states, is 10 times $-0.1(\log_2(0.1)) = 3.32$ bits. Pulling vocal states across all the 228 songs we recorded gives a diversity of 3.24 bits, indicating that abundances of vocal states are distributed fairly uniform in our colony. This could mean that either song diversity tends to be high within subjects, or alternatively, that different song 'morphs' are evenly distributed. In the latter case song diversity would be low within subjects and high when pooled together. Interestingly, the median song diversity of the population was 3.14 bits (**Fig. 4d**), fairly close to the pooled diversity and to the upper theoretical limit,

suggesting a trend to develop acoustically “balanced” songs with respect to the 10 vocal states.”

Line 209: what are "vacant vocal states"?

R3.18 This relates to point R3.17 above: a single bird may (and sometimes do) have near zero abundance of a vocal state.

Line 277: I think you mean "Assuming a natural trend to develop high diversity songs either via imitation or improvisation"??

Fixed

Figure 5: another very busy mega-figure. It's likely not all of this is necessary. 5e is perhaps difficult to interpret and not necessary.

R3.19 We replaced 5e with a new panel demonstrating the new statistical analysis, see point R3.4 above

In a-d, why represent each pair with 10 non-independent points?

R3.20 We think it is important to show these raw data. We overcame the statistical dependency using bootstrap analysis at the tutor-pupil pairs level (see point R3.4). Beyond that, after demonstrating overall statistical significance, it is useful to show the asymmetries in high and low similarity quadrants.

What data were used for the analysis?

R3.20 All the data, using mixed effect model where bird identity is preserved with repeated observations. We now mention it was all data.

Also the figure caption seems to run out of text or the text has been cut off at the end. Subfigure i is indicated as (i) whereas for others it is in bold. In the figure, letters h and i are lost anyway.

Fixed

Figure 6: the caption seems to be truncated.

Fixed

Discussion

Line 452: insert “in” after “abundant”; “pupils” should be “pupil’s “

Fixed

Line 473: insert “to” after “due”

Fixed

Lines 459-460: You call this phenomenon “balanced imitation” but I don’t think that you’ve presented any evidence that the new syllables included by tutors when imitating poor tutors are imitative. Sure, they’re adding new syllables, but where from? Are they imitations or have they developed them de novo, perhaps from some innate template? If they are imitative, who have they imitated? Early in the MS you dismiss that they’re imitating neighbouring tutors although I would find that the most parsimonious explanation. Indeed, dismissing this probably requires more evidence than presented in the MS at this point. Anyway my point is if it really is “balanced imitation” they have to be imitating someone, but who? If not, then it’s a poor term and maybe should be “balanced vocal production” or “balanced song production”?

R3.21 We hope that we the new evidence and analysis on this point is now more solid: i). we now show directly the lack of evidence imitation from neighbors; ii) we now show statistically a bias of deviations (errors) toward balancing; iii) we do show specific examples of balancing in cases where imitation is apparent, even in cases where all tutor syllables were copied. So, we feel that we now have sufficient evidence to use ‘balanced imitation’. Having said that, we do believe that improvisation could be contributing to balancing what is imitated.

Line 503: insert “Australian” before “zebra” at end of sentence.

Fixed

Para 496 – 503: This is very speculative. On the one hand you argue that less variability in songs is due to weaker balanced imitation, but it could also be due to stronger imitation as you point out earlier in the MS.

R3.22. We agree and revised the paragraph. All we want to say is that differences in song diversity across related species could be potentially due to differences in the strength of the tendency of a bird to balance its song.

Lines 519-521: On splitting being rare: as mentioned earlier, this is likely an artefact of your methods. If they split a syllable, you would not have matched it and would not record that whereas when they combine full syllables, they are matched to existing syllables and detected. This needs revision.

R3.23. We can, in principle detect splitting but we agree that it is more challenging. We added a method section to clarify this issue:

“Evaluation of syllable type recombination: Our measure of syllable type diversity is based on classifying syllable types within each bird, and it does not evaluate or compare syllable types across birds. For evaluating syllable type recombination, however, we need to determine which syllables were copied, either as a unit or in parts. Using similarity measurement with Sound Analysis Pro, for each syllable in tutor song we automatically detected sections of similarity in pupil’s song. Evaluating these sections allowed us to determined if the boundaries are consistent across tutor’s and pupil’s songs. In particular

we could detect splitting (a similarity section with one interval in tutor song and two disjoint intervals in pupil's song) or merging (a similarity section with one interval in pupil song and two disjoint intervals in tutor's song). Note, however, that we only included cases where imitation was subjectively apparent, which is easier to determine in cases of merging compared to splitting. Therefore, our lower bound estimates could be biased toward the detection of merging.”

Methods

Lines 564-567: More details required. Measurement of similarity in acoustics is complex and can be done many ways. Just saying it was done using a program with “standard settings” does not provide nearly enough information to have any idea how this was done. What is the underlying method used? What are the relevant settings?

R3.24 The underlying method and relevant settings are now included.

Line 570: Link to Sound Analysis Pro doesn't work. In fact I can't find any working webpage for this software. There is therefore no information about feature measurement in the 10ms windows.

R3.25 Our web server for Sound Analysis Pro was down for a few days; we are sorry about that. It is now working: <http://soundanalysispro.com/manual-1>

Line 572: 50dB relative to what? dB are not absolute units and have little meaning without a reference value.

R3.26 The baseline we use for calculating dB in Sound Analysis Pro is 70dB, which we now indicated.

Line 574: Pitch vs frequency. See notes above about this. If you used frequency then say so; if you logged it first then say this and use pitch.

R3.27 See point R3.14. As to log: we no longer use cepstrum for pitch detection, but use YIN instead (directly on the waveform without FFT). Either way it is a pitch (or fundamental frequency) estimate.

Line 575: There's no heat map in Fig 1b. Assuming you mean the heatmaps in Fig 3, you can't get FM from a single 1 or even 10ms slice – you'd need a greater duration than that. This needs clarification and revision.

R3.28 see point R3.13. As noted, we use spectral derivatives calculated at 10ms windows with multi-tapers, allowing calculation of FM within a single FFT window.

Lines 579-582: I was confused earlier why you used entropy here. Why not use simple proportion of sounds in each vocal state? Isn't that the relevant thing in your analysis? Entropy seems to take this a step further without adding anything. (See note above about being close to maximum entropy and this being a circular argument.)

R3.29 See R3.17 above with respect to argument being circular. We use Shannon entropy to summarize the proportion of vocal states, as our song diversity measure. This holistic measure is needed because it allows us to test whether tutor song diversity affects song imitation. Indeed, the proportion of each vocal states can explain the overall effect, but this is not an entirely trivial outcome. In fact, we initially suspected that it is the overall (holistic) structure of the song that makes some pupils reject those songs. So, we still think that there is merit to looking both holistically and at vocal state abundance separately. Also, as we noted earlier, it is not trivial that entropy is close to the maximum in most bird with respect to the 10 vocal states: one can imagine a population of birds where each bird song is biased (unbalance) in a different manner. E.g., once song with many high pitch syllables, another song with many harmonic syllables, etc. In fact, we do see such cases. Even if all birds were like this, the overall distribution of states could have been similar at the population level with each song being of low diversity. We now look at this quantitatively by comparing the mean song diversity to the pooled song diversity in bottom and top quartiles, see **Figure 7a**;

As shown, low diversity songs, when pooled together, show fairly high diversity. Still, low diversity songs, when pooled together, tend to have less balanced vocals states, with more high pitch song elements (states 9 & 10). This is reminiscent of isolate songs that also often higher in pitch compared to wild type songs.

REVIEWERS' COMMENTS

Reviewer #1 (Remarks to the Author):

The authors fully addressed all my comments. The revised manuscript is excellent and is ready for publication in Nature Communications.

Reviewer #2 (Remarks to the Author):

The authors have made substantial changes, which have greatly improved the paper. I still find the results very interesting and am confident the statistical/computational tools created to analyze song will be of great interest to a large number of researchers. In addition, the findings are novel and important as we think about cultural transmission of vocal gestures. That is, how is diversity maintained with limitations due to imitation.

I have just a few minor comments on the current version of the paper.

1. Line 64: when tutors (not pupils) have low diversity song, correct?
2. It might be useful adding the number of grand-tutors and grand pupils to the first paragraph of the results. It is an important feature of the analysis and comes as a bit of a surprise half way through.
3. Line 153: Is high syllable diversity equivalent to larger 'bits'? If this is correct, then isn't the left tail rare songs with low syllable diversity? Sorry, if I did not read this correctly.
4. Line 84-88. I thought it was a little odd to compare the distribution (Fig1b) with a single number/average – the across families CV. Could you add the average for the within family song similarity CV to the graph?
5. Line 88&89 seem to belong in the next set of questions as it is a separate figure (Figure 2). Can you move this to the next paragraph?
6. On line 223-224 you define song diversity as the proportion of sounds within each vocal state. Then, in the next paragraph (line 236) you refer to vocal state diversities. Is this the same as song diversity?
7. I'm really not sure what line 359-361 means.
8. Could the example pointed out in line 432-434 be illustrated in the graph? I found it difficult to follow the example.
9. For Figure 6. Do you predict the four groups in the pie charts will trend towards 25% of the chart? I think adding some language about how the pie charts demonstrate an increase in balanced imitation would be helpful.

Minor typos

Line 239: diversity explained (not explaining)

Line 365: not more likely than chance

Line 473: last work should be then (not than)

Reviewer #3 (Remarks to the Author):

Review of "Balanced imitation sustains song culture in zebra finches" by Ofer Tchernichovski, Sophie Eisenberg-Edidin and Erich D. Jarvis. This is a revised manuscript. In this paper, the authors use zebra finches to examine details of song learning. They find that pupils tend to compensate for the deficiencies of their tutors. They describe a process they term 'balanced imitation' where rare sounds are preserved and overused sounds become less common in a process that constantly maintains a relatively stable repertoire within a population.

The authors have made significant changes to the manuscript and the manuscript is greatly improved particularly with clarification of some of the details and concepts presented. The authors should be congratulated for addressing the reviewers comments comprehensively.

Response to reviewer #2 comments:

1. Line 64: when tutors (not pupils) have low diversity song, correct?

Yes, fixed.

2. It might be useful adding the number of grand-tutors and grand pupils to the first paragraph of the results. It is an important feature of the analysis and comes as a bit of a surprise half way through.

Done. “We also analyzed song imitation across three generations including 14 grand-tutors and 35 grand-pupils”.

3. Line 153: Is high syllable diversity equivalent to larger ‘bits’? If this is correct, then isn’t the left tail rare songs with low syllable diversity? Sorry, if I did not read this correctly.

Yes, high syllable diversity equivalent to larger bits, and the left tail are rare songs with low syllable diversity. We have now fixed this statement to change it to ‘left tail’ instead of ‘right tail’. It does not change our conclusions.

4. Line 84-88. I thought it was a little odd to compare the distribution (Fig1b) with a single number/average – the across families CV. Could you add the average for the within family song similarity CV to the graph?

Fixed: we added the mean and s.e.m. for CVs in figure legend

5. Line 88&89 seem to belong in the next set of questions as it is a separate figure (Figure 2). Can you move this to the next paragraph?

Done.

6. On line 223-224 you define song diversity as the proportion of sounds within each vocal state. Then, in the next paragraph (line 236) you refer to vocal state diversities. Is this the same as song diversity?

Yes, should be ‘song diversity’. We have fixed it.

7. I’m really not sure what line 359-361 means.

We clarified the sentence: “We noted that in all quartiles, the slope of the correlation was less than one (**Fig. 6a-d**), meaning that when more data tended above the diagonal the abundance of a vocal state was low in a tutor’s song, and when tended below the diagonal the abundance was high in the tutor’s song.”

8. Could the example pointed out in line 432-434 be illustrated in the graph? I found it difficult to follow the example.

We added the example of tutor HP10 in Fig. 5b to the sentence.

9. For Figure 6. Do you predict the four groups in the pie charts will trend towards 25% of the chart? I think adding some language about how the pie charts demonstrate an increase in balanced imitation would be helpful.

Yes, we expect more uniform pie charts in pupils compared their tutors. Clarification added to figure legend. “Note the more uniform pie charts in pupils compared to their tutors”

Minor typos

Line 239: diversity explained (not explaining)

Fixed

Line 365: not more likely than chance

Fixed

Line 473: last work should be then (not than)

Fixed